# ATMAN: Understanding Transformer Predictions Through Memory Efficient Attention Manipulation

**Björn Deiseroth**[1,2,3*]    **Mayukh Deb**[1*]    **Samuel Weinbach**[1*]    **Manuel Brack**[2,4]
**Patrick Schramowski**[2,3,4,5]    **Kristian Kersting**[2,3,4]

[1]Aleph Alpha    [2]Technical University Darmstadt
[3]Hessian Center for Artificial Intelligence (hessian.AI)
[4]German Center for Artificial Intelligence (DFKI)    [5]LAION
{bjoern.deiseroth, mayukh.deb, samuel.weinbach}@aleph-alpha.com
{manuel.brack, patrick.schramowski}@dfki.de
{kersting}@cs.tu-darmstadt.de

## Abstract

Generative transformer models have become increasingly complex, with large numbers of parameters and the ability to process multiple input modalities. Current methods for explaining their predictions are resource-intensive. Most crucially, they require prohibitively large amounts of additional memory since they rely on backpropagation which allocates almost twice as much GPU memory as the forward pass. This renders it difficult, if not impossible, to use explanations in production. We present ATMAN that provides explanations of generative transformer models at almost no extra cost. Specifically, ATMAN is a modality-agnostic perturbation method that manipulates the attention mechanisms of transformers to produce relevance maps for the input with respect to the output prediction. Instead of using backpropagation, ATMAN applies a parallelizable token-based search method relying on cosine similarity neighborhood in the embedding space. Our exhaustive experiments on text and image-text benchmarks demonstrate that ATMAN outperforms current state-of-the-art gradient-based methods on several metrics and models while being computationally efficient. As such, ATMAN is suitable for use in large model inference deployments.

## 1   Explainability through attention maps

Generalizing beyond single-task solutions using large-scale transformer-based language models has gained increasing attention from the community. In particular, the switch to open-vocabulary predictions promises AI systems capable of generalizing to new tasks. Arguably, transformers are the state-of-the-art method in Natural Language Processing (NLP) and Computer Vision. They have shown exceptional performance in multi-modal scenarios, e.g., bridging Computer Vision (CV) capabilities with text understanding to solve Visual Question Answering (VQA) scenarios [10, 18, 31, 30]. However, the increasing adoption of transformers also raises the need to understand their otherwise black-box predictions better. Unfortunately, the "scale is all you need" assumption of recent transformer models results in severely large and complex architectures. This renders tasks such as their training, inference deployment, and explaining of their outputs resource-intensive, requiring multiple enterprise-grade GPUs or even entire computing nodes, along with prolonged runtimes.

---

[*] These authors contributed equally to this work
Source code: https://github.com/Aleph-Alpha/AtMan
Ātman (Sanskrit) "essence"

37th Conference on Neural Information Processing Systems (NeurIPS 2023).

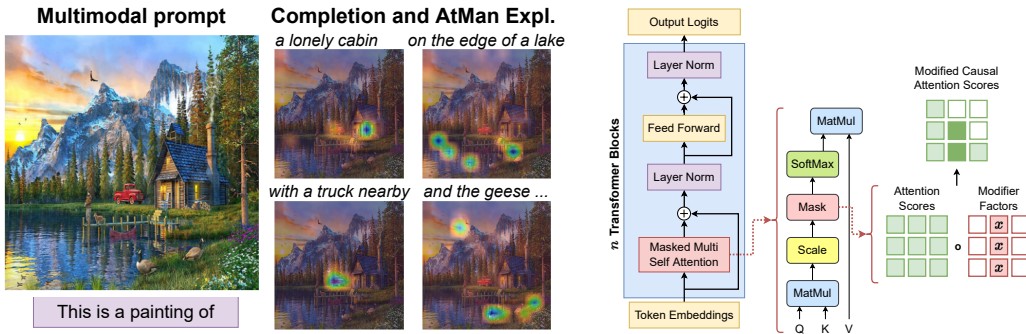

(a) "What am I looking at?"      (b) ATMAN in the transformer architecture.

Figure 1: **(a)** The proposed explainability method ATMAN visualizes the most important aspects of the given image while completing the sequence, displayed above the relevance maps. The generative multi-modal model MAGMA is prompted to describe the shown image with: "<Image> This is a painting of". **(b)** The integration of ATMAN into the transformer architecture. We multiply the modifier factors and the attention scores before applying the diagonal causal attention mask as depicted on the right-hand side. Red hollow boxes (☐) indicate one-values, and green ones (☐) -infinity. (Best viewed in color.)

Explainable AI (XAI) methods target to elucidate the decision-making processes and internal workings of AI models to humans. State-of-the-art methods for transformers focus on propagating gradients back through the model. This backpropagation allows for the accumulation of influence of each input feature on the output by utilizing stored activations during the forward pass [6, 27]. Unfortunately, this caching also leads to significant overhead in memory consumption, which renders their productive deployment to be uneconomical, if not impossible. Often half of the available memory of the GPU has to remain unused at inference, or it requires an entirely separate deployment of the XAI pipeline.

Fortunately, another popular XAI idea, namely *perturbation* [19, 24], is much more memory-efficient. However, it has not yet been proven beneficial for explaining the predictions of transformers, as the immense number of necessary forward trials to derive the explanation accumulate impractical computation time.

To tackle these issues and in turn, scale explanations with the size of transformers, we propose to bridge relevance propagation and perturbations. In contrast to existing perturbation methods— executing perturbations directly in the input space—ATMAN pushes the perturbation steps via **at**tention **man**ipulations throughout the latent layers of the transformer during the forward pass. This enables us—as we will show—to produce state interpolations, token-based similarity measures and to accurately steer the model predictions. Our explanation method leverages these predictions to compute relevancy values for transformer networks. Our experimental results demonstrate that ATMAN significantly reduces the number of required perturbations, making them applicable at deployment time, and does not require additional memory compared to backpropagation methods. In short, ATMAN can scale with transformers. Our exhaustive experiments on the text and image-text benchmarks and models demonstrate that ATMAN outperforms current state-of-the-art gradient-based methods while being more computationally efficient. For the first time, ATMAN allows one to study generative model predictions as visualized in Fig. 1a without extra deployment costs. During the sequence generation with large multi-modal models, ATMAN is able to additionally highlight relevant features wrt. the input providing novel insights on the generation process.

**Contributions.** In summary, our contributions are: (i) An examination of the effects of token-based attention score manipulation on generative transformer models. (ii) The introduction of a novel and memory-efficient XAI perturbation method for large-scale transformer models, called ATMAN, which reduces the number of required iterations to a computable amount by correlating tokens in the embedding space. (iii) Exhaustive multi-modal evaluations of XAI methods on several text and image-text benchmarks and autoregressive (AR) transformers.

We proceed as follows. We start by discussing related work in Sec. 2. In Sec. 3, we derive ATMAN and explain its attention manipulation as a perturbation technique. Before concluding and discussing the benefits as well as limitations in Sec. 5, we touch upon our experimental evaluation Sec. 4,

showing that ATMAN not only nullifies memory overhead and proves robust to other models but also outperforms competitors on several visual and textual reasoning benchmarks.

## 2 Related Work

**Explainability in CV and NLP.** The explainability of AI systems is still ambiguously defined [8]. XAI methods are expected to show some level of relevance to the input with respect to the computed result of an algorithm. This task is usually tackled by constructing an input relevance map given the model's prediction. The nature of relevance can be class-specific, e.g., depending on specific target instances of a task and showing a "local solution" [26, 27], or class-agnostic, i.e., depending on the overall "global behavior" of the model only [1, 4]. The level of fine granularity in the achieved explanation thus depends on the selected method, model, and evaluation benchmark. Explainability in CV is usually achieved by mapping the relevance maps to a pixel level and treating the evaluation as a weak segmentation task [25, 20, 27]. On the other hand, NLP explanations are much more vaguely defined and usually mixed with more complex philosophical interpretations, such as labeling a given text to a certain sentiment category [8].

The majority of XAI methods can be divided into the classes of perturbation and gradient analysis. Perturbations treat the model as a black box and attempt to derive knowledge of the model's behavior by studying changes in input-output pairs only. Gradient-based methods, on the other hand, execute a backpropagation step towards a target and aggregate the model's parameter adoptions to derive insights. Most of these XAI methods are not motivated by a specific discipline, e.g., neither by NLP nor CV. They are so generic that they can be applied to both disciplines to some extent. However, architecture-specific XAI methods exist as well, such as GradCAM [25], leveraging convolutional neural networks' spatial input aggregation in the deepest layers to increase efficiency.

**Explainability in transformers.** Through their increasing size, transformers are particularly challenging for explainability methods, especially for architecture-agnostic ones. Transformers' core components include an embedding layer followed by multiple layers of alternating attention and feed-forward blocks Fig. 1b. The attention blocks map the input into separate "query", "key", and "value" matrices and are split into an array of "heads". As with convolutions in CNN networks, separated heads are believed to relate to specific learned features or tasks [13]. Further, the attention mechanism correlates tokens wrt. the input sequence dimension. Care needs to be taken here as GPT-style generative models, in contrast to BERT-style embedding models, apply a causal masking of activations as we will highlight further below. This in particular might obscure results for gradient-based methods.

Consequently, most explainability adoptions to transformers focus on the attention mechanism. Rollout [1] assumes that activations in attention layers are combined linearly and consider paths along the pairwise attention graph. However, while being efficient, it often emphasizes irrelevant tokens, in particular, due to its class-agnostic nature. The authors also propose attention flow, which, however, is unfeasible to compute as it constructs exhaustive graphs. More recently, Chefer et al. (2021) proposed to aggregate backward gradients and LRP [20] throughout all layers and heads of the attention modules in order to derive explanation relevancy. Their introduced method outperforms previous transformer-specific and unspecific XAI methods on several benchmarks and transformer models. It is further extended to multimodal transformers [7] by studying other variations of attention. However, they evaluated only on classification tasks, despite autoregressive transformers' remarkable generative performance, e.g., utilizing InstructGPT [21] or multimodal transformers such as MAGMA [10], BLIP [18] and OFA [31].

**Multimodal transformers.** Contrarily to these explainability studies evaluated on object-detection models like DETR and ViT [5, 9], we study explainability on open-vocabulary tasks, i.p., generated text tokens of a language model, and not specifically trained classifiers. Due to the multimodality, the XAI method should produce output relevancy either on the input text or the input image as depicted in Fig. 1a. Specifically, to obtain image modality in MAGMA [10], the authors propose to fine-tune a frozen pre-trained language model by adding sequential adapters to the layers, leaving the attention mechanism untouched. It uses a CLIP [22] vision encoder to produce image embeddings. These embeddings are afterward treated as equal during model execution, particularly regarding

other modality input tokens. This multi-modal open-vocabulary methodology has shown competitive performance compared to single-task solutions.

# 3 ATMAN: Attention Manipulation

We formulate finding the best explainability estimator of a model as solving the following question: *What is the most important part of the input, annotated by the explanator, to produce the model's output?* In the following, we derive our perturbation probe mathematically through studies of influence functions and embedding layer updates on autoregressive (AR) models [14, 2].

Then we show how attention manipulation on single tokens can be used in NLP tasks to steer a model's prediction in directions found within the prompt. Finally, we derive our multi-modal XAI method ATMAN by extending this concept to the cosine neighborhood in the embedding space.

## 3.1 Influence functions as explainability estimators

Transformer-based language models are probability distribution estimators. They map from some input space $\mathcal{X}$, e.g., text or image embeddings, to an output space $\mathcal{Y}$, e.g., language token probabilities. Let $\mathcal{E}$ be the space of all explanations, i.e., binary labels, over $\mathcal{X}$. An explanator function can then be defined as $e : (\mathcal{X} \to \mathcal{Y}) \times (\mathcal{X} \times \mathcal{Y}) \to \mathcal{E}$, in words, given a model, an input, and a *target*, derive a label on the input.

Given a sequence of words $\mathbf{w} = [w_1, \ldots, w_N] \in \mathcal{X}^N$, an AR language model assigns a probability to that sequence $p(\mathbf{w})$ by applying factorization $p(\mathbf{w}) = \prod_t p(w_t|w_{<t})$. The loss optimization during training can then be formalized as solving:

$$\max_\theta \log p_\theta(\mathbf{w})_{target} = \sum_t \log p_\theta(w_t|w_{<t})_{target^t} = \sum_t \log \text{softmax}(h_\theta(w_{<t})W_\theta^T)_{target^t} \quad (1)$$

$$=: -\sum_t L^{target}(\mathbf{w}_{<t}, \theta) =: -L^{target}(\mathbf{w}, \theta) . \quad (2)$$

Here $h_\theta$ denotes the model, $W_\theta$ the learned embedding matrix, and *target^t* the vocabulary index of the $t - th$ target token, with length $|target| = N$. Eq. 1 is derived by integrating the cross-entropy loss, commonly used during language model training with *target* = $\mathbf{w}$. Finally, $L$ denotes our loss function.

Perturbation methods study the influence of the model's predictions by adding small noise $\epsilon$ to the input and measuring the prediction change. We follow the results of the studies [14, 2] to approximate the perturbation effect directly through the model's parameters when executing Leaving-One-Out experiments on the input. The influence function estimating the perturbation $\epsilon$ of an input $z$ is then derived as:

$$\mathcal{I}^{target}(z_\epsilon, z) = \frac{dL^{target}(z, \theta_\epsilon)}{d\epsilon}\Big|_{\epsilon=0} \approx L^{target}(z, \theta_{-z_\epsilon}) - L^{target}(z, \theta) . \quad (3)$$

Here $\theta_{-z_\epsilon}$ denotes the set of model parameters in which $z_\epsilon$ would not have been seen during training. In the following, we further show how to approximate $\theta_{-z_\epsilon}$.

## 3.2 Single token attention manipulation

The core idea of ATMAN is the shift of the perturbation space from the raw input to the embedded token space. This allows us to reduce the dimensionality of possible perturbations down to a single scaling factor per input token. Moreover, we do not manipulate the value matrix of attention blocks and therewith do not introduce the otherwise inherent input-distribution shift of obfuscation methods. By manipulating the attention scores at the positions of the corresponding input sequence tokens, we are able to interpolate the focus of the prediction distribution of the model—amplifying or suppressing concepts of the prompt resulting in an XAI method as described in the following.

Attention was introduced in [28] as: $\mathbf{O} = \text{softmax}(\mathbf{H}) \cdot \mathbf{V}$, where $\cdot$ denotes matrix multiplication. The pre-softmax query-key attention scores are defined as

$$\mathbf{H} = \mathbf{Q} \cdot \mathbf{K}^T / \sqrt{d_h}.$$

In the case of autoregression, a lower left triangular unit mask $\mathbf{M}$ is applied to these scores as $\mathbf{H}_M = \mathbf{H} \circ \mathbf{M}$, with $\circ$ the Hadamard product. The output of the self-attention module is $\mathbf{O} \in$

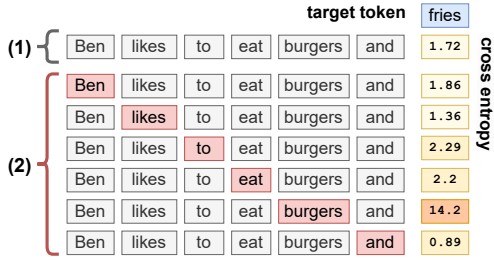

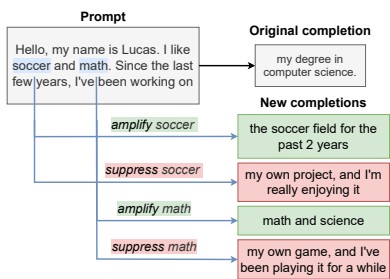

(a) Illustration of the perturbation method.    (b) Steering through AtMan.

Figure 2: **(a) Illustration of the proposed explainability method.** First, we collect the original cross-entropy score of the target tokens (1). Then we iterate and suppress one token at a time, indicated by the red box (☐), and track changes in the cross-entropy score of the target token (2). **(b) Manipulation of the attention scores**, highlighted in blue, **steers the model's prediction** into a different contextual direction. Note that we found measuring of such proper generative directions to perform better than radical token-masking, as we show later (c.f. Fig. 11). (Best viewed in color.)

$\mathbb{R}^{h \times s \times d_h}$, the query matrix is $\mathbf{Q} \in \mathbb{R}^{h \times s \times d_h}$ and $\mathbf{K}, \mathbf{V} \in \mathbb{R}^{h \times s \times d_h}$ the keys and values matrices. Finally $\mathbf{H}, \mathbf{M}, \mathbf{H}_M \in \mathbb{R}^{h \times s \times s}$. The number of heads is denoted as $h$, and $d_h$ is the embedding dimension of the model. Finally, $s$ is the length input-sequence.

The perturbation approximation $\theta_{-z_\epsilon}$ required by Sec. 3.1 can now be approximated through attention score manipulation as follows: Let $\mathbf{w}$ be an input token sequence of length $|\mathbf{w}| = n$. Let $i$ be a token index within this sequence to be perturbated by a factor $f$. For all layers and all heads $\mathbf{H}_u$ we modify the pre-softmax attention scores as:

$$\widetilde{\mathbf{H}}_{u,*,*} = \mathbf{H}_{u,*,*} \circ (\mathbf{1} - \mathbf{f}^i), \tag{4}$$

where $\mathbf{1} \in [1]^{s \times s}$ denotes the matrix containing only ones and $\mathbf{f}^i$ the suppression factor matrix for token $i$. In this section we set $\mathbf{f}^i_{k,*} = f$, for $k = i$ and $f \in \mathbb{R}$ and 0 elsewhere. As depicted in Fig. 1b, we thus only amplify the $i-th$ column of the attention scores of $H$ by a factor $(1 - f)$—this, however, equally for all heads. We denote this modification to the model as $\theta_{-i}$ and assume a fixed factor $f$.[1] Note that this position in the architecture is somewhat predestined for this kind of modification, as it already applies the previously mentioned causal token-masking.

We define the explanation for a class label *target* as the vector of the positional influence functions:

$$\mathcal{E}(\mathbf{w}, target) \coloneqq \left( \mathcal{I}^{target}(\mathbf{w}_1, \mathbf{w}), \dots, \mathcal{I}^{target}(\mathbf{w}_n, \mathbf{w}) \right), \tag{5}$$

with $\mathcal{I}^{target}$ derived by Eq. 2,3 as $\mathcal{I}^{target}(\mathbf{w}_i, \mathbf{w}) \coloneqq L^{target}(\mathbf{w}, \theta_{-i}) - L^{target}(\mathbf{w}, \theta)$. In words, we average the cross-entropy of the AR input sequence wrt. all target tokens and measure the change when suppressing token index $i$ to the unmodified one. The explanation becomes this difference vector of all possible sequence position perturbations and thus requires $n$ forward passes.

Fig. 2a illustrates this algorithm. The original input prompt is the text "Ben likes to eat burgers and" for which we want to extract the most valuable token for the completion of the target token "fries". Initially, the model predicts the target token with a cross-entropy score of $1.72$. Iterated suppression over the input tokens, as described, leads to "burgers" being the most-influential input token with the highest score of $14.2$ for the completion.

**Token attention suppression steers the model's prediction.** Intuitively, for factors $0 < f < 1$, we call the modifications "suppression", as we find the model's output now relatively less influenced by the token at the position of the respective manipulated attention scores. Contrarily, $f < 0$ "amplifies" the influence of the manipulated input token on the output.

An example of the varying continuations, when a single token manipulation is applied, can be seen in Fig. 2b. We provide the model a prompt in which the focus of continuation largely depends on two tokens, namely "soccer" and "math". We show how suppressing and amplifying them alters the prediction distributions away from or towards those concepts. It is precisely this distribution shift we measure and visualize as our explainability.

---

[1]We ran a parameter sweep once (c.f. App. A.4) and fixed this parameter to 0.9 for this work.

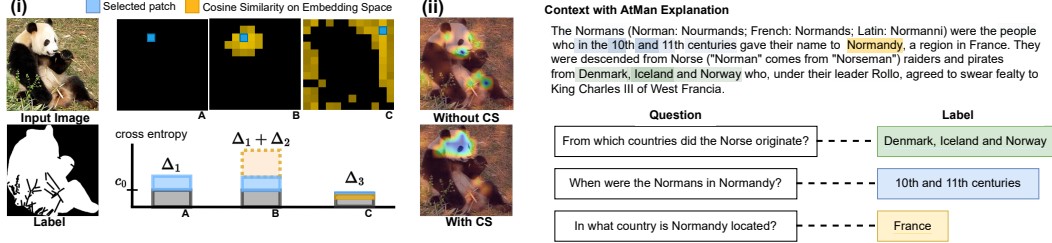

(a) Correlated token suppression on images.      (b) SQuAD example with ATMAN Explanations.

Figure 3: **(a) Correlated token suppression of ATMAN enhances explainability in the image domain.** i) Shows an input image along with three perturbation examples $(A, B, C)$. In $A$, we only suppress a single image token (blue). In $B$, the same token with its relative cosine neighborhood (yellow), and in $C$, a non-related token with its neighborhood. Below depicted are the changes in the cross-entropy loss. The original score for the target token "panda" is denoted by $c_0$ and the loss change by $\Delta$. ii) Shows the resulting explanation without Cosine Similarity (CS) and with CS. We evaluated the influence of the CS quantitatively in Fig. 12a. **(b)** An example of **the SQuAD dataset with ATMAN explanations**. The instance contains three questions for a given context, each with a labeled answer pointing to a fragment of the context. ATMAN is used to highlight the corresponding fragments of the text responsible for the answer. It can be observed that the green example is full, the blue in part, and the yellow is not at all recovered according to the given labels. However, the yellow highlight seems at least related to the label. (Best viewed in color.)

## 3.3   Correlated token attention manipulation

Suppressing single tokens works well when the entire entropy responsible for producing the target token occurs only once. However, for inputs with redundant information, this approach would often fail. This issue is especially prominent in Computer Vision (CV), where information like objects in an image is often spread across multiple tokens—as the image is usually divided into patches, each of which is separately embedded. It is a common finding that applied cosine similarity in the embedding space, i.p., for CLIP-like encodings as used in MAGMA, gives a good correlation estimator [17, 2]. This is in particular due to the training by a contrastive loss. We integrate this finding into ATMAN in order to suppress all redundant information corresponding to a particular input token at once, which we refer to as correlated token suppression.

Fig. 3a summarizes the correlated token suppression visually. For $n$ input tokens and embedding dimension $d$, the embedded tokens result in a matrix $T = (t_i) \in \mathbb{R}^{n \times d}$. The cosine similarity is computed from the normalized embeddings $\widetilde{T} = (\widetilde{t_i})$, with $\widetilde{t_i} = t_i / \|t_i\|$, for $i \in 1, \ldots n$, as $S = (s_i) = \widetilde{T} \cdot \widetilde{T}^\top \in [-1, 1]^{n \times n}$. Note that the index $i$ denotes a column corresponding to the respective input token index. Intuitively, the vector $s_i$ then contains similarity scores to all input tokens. In order to suppress the correlated neighborhood of a specific token with the index $i$, we, therefore, adjust the suppression factor matrix for Eq. 4 as

$$\widetilde{\mathbf{H}}_{u,*,*} = \mathbf{H}_{u,*,*} \circ \big((\mathbf{1} - f) + f(\mathbf{1} - \mathbf{f}^i_{k,*})\big), \text{ with } \mathbf{f}^i_{k,*} = \begin{cases} s_{i,k}, & \text{if } \kappa \le s_{i,k} \le 1, \\ 0, & \text{otherwise.} \end{cases} \tag{6}$$

In words, we interpolate linear between the given *suppression factor* $1 - f$ and $1$, by factor of the thresholded cosine similarity $\mathbf{f}^i_{k,*}$. As we only want to suppress tokens, we restrict the factor value range to greater than 0 and smaller than 1. The parameter $\kappa > 0$[2] is to ensure a lower bound on the similarity, and in particular, prevents a flip of the sign. Overall, we found this to be the most intuitive and robust equation. In Fig. 11 we ran a sweep over the hyper-parameters to obtain the best precision and recall values. Note in particular, that our modifications are rather subtle compared to entire token-masking. The latter did not lead to satisfying results, even given the context. We compute the similarity scores once on the initial embedding layer, and uniformly apply those modifications throughout all layers.

Note that variations of Eq. 4, 5 and 6 could also lead to promising results. In particular in the active research field of vision transformers, hierarchical token transformations could be applicable and speed

---

[2]We empirically fixed $\kappa = 0.7$ through a parameter sweep (c.f. Appendix A.4).

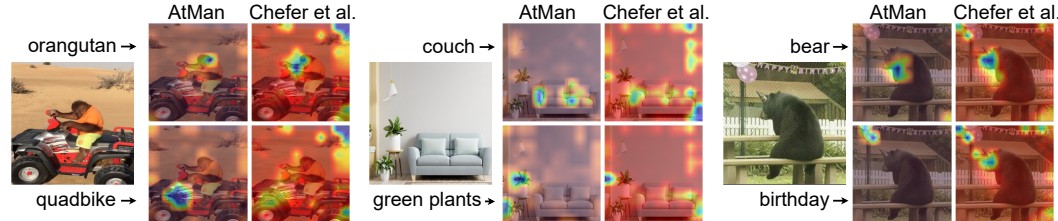

Figure 4: **ATMAN produces less noisy and more focused explanations when prompted with multi-class weak segmentation** compared to Chefer. The three shown figures are prompted to explain the target classes above and below separately. (Best viewed in color.)

up computations (c.f. [3], Fig 16). Furthermore, preliminary experiments using the similarity between the query and key values per layer, a different similarity score such as other embedding models or directly sigmoid values, or even a non-uniform application of the manipulations yield promising, yet degenerated performance (c.f. App. A.15). As we could not conclude a final conclusion yet, we leave these investigations to future research.

With that, we arrived at our final version of ATMAN. Note that this form of explanation $\mathcal{E}$ is "local", as *target* refers to our target class. We can, however, straightforwardly derive a "global explanation" by setting *target* = $\mathbf{y}$, for $\mathbf{y}$, a model completion to input $\mathbf{w}$ of a certain length. It could then be interpreted rather abstract as the model's general focus [4].

## 4    Empirical Evaluation

We ran empirical evaluations on text and image corpora to address the following questions: **(Q1)** Does ATMAN achieve competitive results compared to previous XAI methods for transformers in the language as well as vision domain? **(Q2)** Is ATMAN applicable to various transformer models, and does ATMAN scale efficiently, and can therefore be applied to current large-scale AR models?

To answer these questions, we conducted empirical studies on textual and visual XAI benchmarks and compared ATMAN to standard approaches such as IxG [26], IG [27], GradCAM [25] and the transformer-specific XAI method of [6] called Chefer in the following. Note that all these methods utilize gradients and, therefore, categorize as propagation methods leading to memory inefficiency. We also tried to apply existing perturbation methods such as LIME [24] and SHAP [19] and Attention Rollout [1]. However, they failed due to extremely large trials and, in turn, prohibitive computation time, or qualitatively (c.f. Appendix A.9). We adopt common metrics, namely mean average precision (mAP) and recall (mAR) (c.f. Appendix A.11), and state their interquartile statistics in all experiments. To provide a comparison between the XAI methods, we benchmarked all methods on GPT-J [29] for language and MAGMA-6B [10] for vision-language tasks. Through its memory efficiency, ATMAN can be deployed on really large models. We further evaluate ATMAN on a MAGMA-13B and 30B to show the scaling behavior, and, on BLIP [18] to show the architecture independence. To obtain the most balanced comparison between the methods, we selected datasets focusing on pure information recall, separating the XAI evaluation from the underlying model interpretations.

### 4.1    ATMAN can do language reasoning

**Protocol.**    All experiments were performed on the open-source model GPT-J [29]. Since with ATMAN we aim to study large-scale generative models, we formulate XAI on generative tasks as described in Sec. 3.3. For evaluation, we used the Stanford Question Answering (QA) Dataset (SQuAD) [23]. The QA dataset is structured as follows: Given a single paragraph of information, there are multiple questions, each with a corresponding answer referring to a position in the given paragraph. A visualization of an instance of this dataset can be found in Fig. 3b. SQuAD contains 536 unique paragraphs and 107,785 question/explanation pairs. The average context sequence length is 152.7 tokens and the average label length is 3.4.

The model was prompted with the template: "{Context} Q: {Question} A:", and the explainability methods evaluated on the generations to match the tokens inside the given context, c.f. Fig. 3b, Sec. 3.3. If there were multiple tokens in the target label, we computed the mean of the generated scores.

Table 1: **ATMAN (AM) outperforms other XAI methods** on **(a)** the text QA benchmark SQuAD and **(b)** the image-text VQA task of OpenImages. Shown are the (interquartile) mean average precision and recall (the higher, the better). The best and second best values are highlighted with ● and ○. Evaluations are on a 6B model. **(c)** ATMAN evaluated on other model types: MAGMA variants (indicated by their parameter size) and BLIP.

| | (a) SQuAD. | | | | (b) OpenImages (OI). | | | | | (c) Other models on OI. | | |
|---|---|---|---|---|---|---|---|---|---|---|---|---|
| | IG | IxG | Chef. | AM | GC | IxG | IG | Chef. | AM | $AM_{13B}$ | $AM_{30B}$ | $AM_{BLIP}$ |
| mAP | 49.5 | 51.7 | ○72.7 | ●73.7 | 55.2 | 59.1 | ○60.5 | 58.3 | ●65.5 | 64.8 | 63.6 | 64.6 |
| $mAP_{IQ}$ | 49.5 | 61.4 | ○77.5 | ●81.8 | 54.3 | 59.0 | ○60.6 | 57.8 | ●70.2 | 65.6 | 65.1 | 66.1 |
| mAR | 87.1 | 91.8 | ●96.6 | ○93.4 | 4.2 | 10.8 | 10.7 | ○11.7 | ●15.7 | 14.6 | 13.6 | 26.4 |
| $mAR_{IQ}$ | 98.6 | ●100 | ●100 | ●100 | 3.8 | 10.2 | 10.0 | ○11.1 | ●19.7 | 16.7 | 15.6 | 28.4 |

Similar to weak segmentation tasks in computer vision, we regarded the annotated explanations as binary labels and determined precision and recall over all these target tokens.

**Results.** The results are shown in Tab. 1a. It can be observed that the proposed ATMAN method thoroughly outperforms all previous approaches on the mean average precision. This statement holds as well for the interquartile mean of the recall. However on the entire recall, Chefer slightly outperforms ATMAN. Note that the high values for recall scores throughout all methods are partly due to the small average explanation length, such as depicted in Fig. 3b. Further details and some qualitative examples can be found in Appendix A.2.

**Paragraph chunking.** ATMAN can naturally be lifted to the explanation of paragraphs. We ran experiments for ATMAN dividing the input text into several paragraphs by using common delimiters as separation points and evaluating the resulting chunks at once, in contrast to token-wise evaluations. This significantly decreases the total number of required forward passes and, on top, produces "more human" text explanations of the otherwise still heterogeneously highlighted word parts. Results are shown in Appendix A.8.

## 4.2 ATMAN can do visual reasoning

**Protocol.** Similar to language reasoning, we again perform XAI on generative models. We evaluated the OpenImages [16] dataset as a VQA task and generated open-vocabulary prediction with the autoregressive model. Specifically, the model is prompted with the template: "{Image} This is a picture of ", and the explainability methods executed to derive scores for the pixels of the image on the generations with respect to the target tokens. If there were multiple tokens in the target label, we computed the mean of the generated scores. For evaluation, we considered the segmentation annotations of the dataset as ground truth explanations. The segmentation subset contains 2,7M annotated images for 350 different classes. In order to ensure a good performance of the large-scale model at hand and, in turn, adequately evaluate only the XAI methods, we filtered the images for a minimum dimension of 200 pixels and a maximal proportional deviation between width and height of 20%. We randomly sample 200 images per class on the filtered set to further reduce evaluation costs. In total, this leads to a dataset of 27.871 samples. The average context sequence length is 144 tokens, and the average label coverage is 56% of the input image. To provide a fair comparison between all XAI methods, we compute token-based attributions for both ATMAN and Chefer, as well as IG, IxG, and GC. Note that the latter three allow pixel-based attribution granularity, however, these resulted in lower performance, as further discussed in Appendix A.2.

**Results.** The results are shown in Tab. 1b. It can be observed that ATMAN thoroughly outperforms all other XAI approaches on the visual reasoning task for all metrics. Note how explicit transformer XAI methods (ATMAN, Chefer) in particular outperform generic methods (GradCAM, IG, IxG) in recall. Moreover, while being memory-efficient (c.f. next section), ATMAN also generates more accurate explanations compared to Chefer. Through the memory efficiency of ATMAN, we were able to evaluate a 13B and 30B upscaled variant of MAGMA (c.f. Tab. 1c). Interestingly, the general explanation performance slightly decreases, when upscaled, compared to the 6B model variant. However, throughout all variants ATMAN scored comparable high. Note that the overall benchmark performances of the models slightly increase (c.f. App. A.14). This behavior could be attributed to

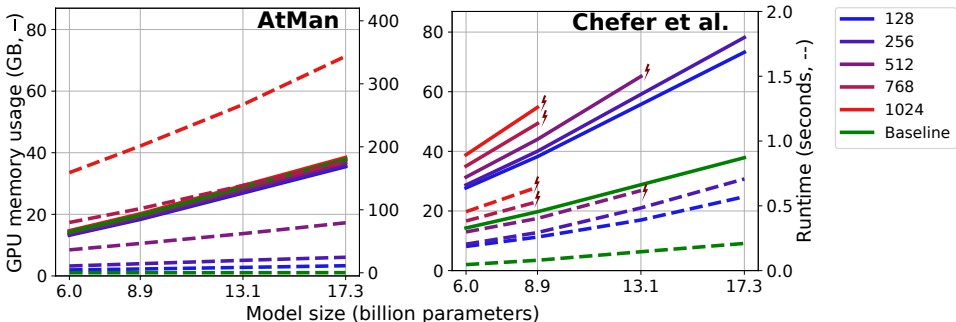

Figure 5: **ATMAN scales efficiently.** Performance comparison of the XAI methods ATMAN and Chefer *et al.*, on various model sizes (x-axis) executed on a single 80GB memory GPU. Current gradient-based approaches do not scale; only ATMAN can be utilized on large-scale models. Solid lines refer to the GPU memory consumption in GB (left y-axis). Dashed lines refer to the runtime in seconds (right y-axis). Colors indicate experiments on varying input sequence lengths. As baseline (green) a plain forward pass with a sequence length of 1024 is measured. The lightning symbol emphasizes the non-deployability when memory resource is capped to 80GB. (Best viewed in color.)

the increased complexity of the model and, subsequently, the complexity of the explanation at hand. Hence, the "human alignment" with the model's explanations is not expected to scale with their size.

**Model agnosticism.** Note that the underlying MAGMA model introduces adapters on a pre-trained decoder text model to obtain the image multi-modality. It is yet unclear to what extent adapters change the flow of information in the underlying model. This may be a reason for the degenerated performance of gradient-based models. As ATMAN however outperforms such methods, we may conclude that information processing still remains bound to the specific input-sequence position. This finding generalizes to encoder architectures, as experiments with BLIP suggest in Tab. 1c. Precisely, BLIP's multi-modality is achieved by combining an image-encoder, a text-encoder, and a final text-decoder model. These are responsible for encoding the image, the question, and finally for generating a response separately. Information between the models is passed through cross-attention. We only applied ATMAN to the image-encoder model of this architecture (c.f. App. A.10). The results demonstrate the robustness of the suggested perturbation method over gradient-based approaches.

**Qualitative illustration.** Fig. 4 shows several generated image explanations of ATMAN and Chefer for different concepts. More examples of all methods can be found in Appendix A.7 and evaluation on the more complex GQA dataset [12] is discussed in Appendix A.12. We generally observe more noise in gradient-based methods, in particular around the edges. Note that ATMAN evaluates the change in its perturbated generations. These are independent of the target tokens. Compared to other methods like Chefer, we, therefore, do not need to process the prompt more than once for VQA on different object classes with ATMAN. Note again that the aforementioned cosine similarity is crucial to obtain these scores. Quantitative results can be found in App. A.4.

In general, the results clearly provide an affirmative answer to **(Q1)**: ATMAN is competitive with previous XAI methods, and even outperforms transformer-specific ones on the evaluated language and vision benchmarks for precision and recall. Next, we will analyze the computational efficiency of ATMAN.

### 4.3 ATMAN can do large scale

While ATMAN shows competitive performance, it computes, unlike previous approaches, explanations at almost no extra memory cost. Fig. 5 illustrates the runtime and memory consumption on a single NVIDIA A100 80GB GPU. We evaluated the gradient-based transformer XAI method [6] and ATMAN. The statistics vary in sequence lengths (colors) from 128 to 1024 tokens, and all experiments are executed sequentially with batch size 1 for best comparison.

One can observe that the memory consumption of ATMAN is around that of a plain forward pass (Baseline; green) and increases only marginally over the sequence lengths. In comparison, the method of [6]—and other gradient-based methods—exceeds the memory limit with more than double in memory consumption. Therefore, they fail on larger sequence lengths or models.

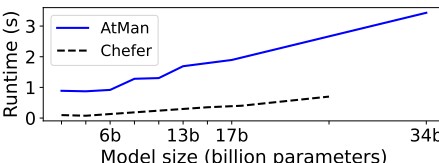 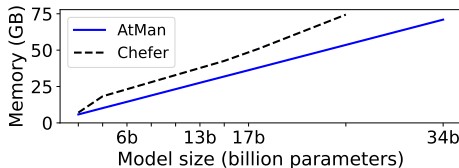

Figure 6: Examplified average runtime and total memory consumption for the task of explaining an image, comparing AtMan with Chefer et al.. We applied memory improvements on Chefer et al., pipe-parallel of 4 on both and additionally batch-size 16 on ATMAN. Chefer et al. still is not applicable to 34b models when the total memory consumption is capped to 80GB. AtMan's increase in total computational time lies around 1 magnitude. Note that AtMan could be computed entirely parallel, and as such, this total clock time can still be reduced to subseconds. Measured are 144 explanation tokens (full given image on MAGMA), for an average total prompt length of 165 tokens.

Whereas the memory consumption of ATMAN stays is almost constant, the execution time significantly increases over sequence length when no further token aggregation is applied upfront. Note, that Fig. 5 shows complete and sequential execution. The exhaustive search loop of ATMAN can be run partially and in parallel to decrease its runtime. In particular, the parallelism can be achieved by increasing the batch size and naturally by a pipeline-parallel execution. For instance, since large models beyond 100B are scattered among nodes and thus many GPUs, the effective runtime is reduced by magnitudes to a proximate scale of the forward pass. Furthermore, tokens can be aggregated in chunks and evaluated on a more coarse level, which further reduces the runtime drastically, as demonstrated in Fig. 16.

In Fig. 6 we provide performance results on a realistic scenario. It compares a practical query (average 165 tokens) on the task of explaining a complete image (144) tokens. Runtimes are measured with pipe-parallel 4, and ATMAN additionally with batch size 16. Specifically, ATMAN requires between 1 and 3 total compute seconds, around 1 magnitude longer compared to Chefer. Note that this can still further be divided by the number of available idling workers, to reduce the absolute clock time to subseconds. Each batch of ATMAN can be processed entirely parallel. Moreover Chefer, even with memory optimizations, fails to scale to 34b with the given memory limit (80GB).

Overall, these results clearly provide an affirmative answer to **(Q2)**: Through the memory efficiency of ATMAN, it can be applied to large-scale transformer-based models.

## 5 Conclusion

We proposed ATMAN, a modality and architecture agnostic perturbation-based XAI method for generative transformer networks. In particular, ATMAN reduces the complex issue of finding proper input perturbations to a single scaling factor per token. As our experiments demonstrate, ATMAN outperforms current approaches relying on gradient computation. It moreover performs well on decoder as well as encoder architectures, even with further adjustments like adapters or crossattention in place. Most importantly, through ATMAN's memory efficiency, it provides the first XAI tool for deployed large-scale AR transformers. We successfully evaluated a 30B multi-modal model on a large text-image corpus.

Consequently, ATMAN paves the way to evaluate whether models' explanatory capabilities scale with their size, which should be studied further. Besides this, our paper provides several avenues for future work, including explanatory studies of current generative models impacting our society. Furthermore, it could lay the foundation for not only instructing and, in turn, improving the predictive outcome of autoregressive models based on human feedback [21] but also their explanations [11].

## Acknowledgments

We gratefully acknowledge support by the German Center for Artificial Intelligence (DFKI) project "SAINT", the Federal Ministry of Education and Research (BMBF) project "AISC" (GA No. 01IS22091), and the Hessian Ministry for Digital Strategy and Development (HMinD) project "AI Innovationlab" (GA No. S-DIW04/0013/003). This work also benefited from the ICT-48 Network

of AI Research Excellence Center "TAILOR" (EU Horizon 2020, GA No 952215), the Hessian Ministry of Higher Education, and the Research and the Arts (HMWK) cluster projects "The Adaptive Mind" and "The Third Wave of AI", and the HMWK and BMBF ATHENE project "AVSV". Further, we thank Felix Friedrich, Marco Bellagente and Constantin Eichenberg for their valuable feedback.

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

# A    Appendix

## A.1    Remarks on executed benchmarks

We executed all benchmarks faithfully and to the best of our knowledge. The selection of compared methods was made to be rather diverse and obtain a good overview in this field of research. In particular, with regards to the multi-modal transformer scaling behavior, as there are in fact no such studies for AR models yet to compare to. It is possible, for all methods, that there are still improvements we missed in quality as well as performance. However, we see the optimizations of other methods to multi-modal AR transformer models as a research direction on its own.

**Chefer.**    The integration of Chefer was straightforward. As it can be derived by the visualizations, there are noticeable artifacts, particularly on the edges of images. In this work the underlying transformer model was MAGMA, which is finetuned using sequential adapters. It is possible that this, or the multi-modal AR nature itself, is the cause for these artifacts. We did not further investigate to what extent the adapters are to be particularly integrated in the attribute accumulation of Chefer. Also notice that ATMAN often has similar, but not as severe, artifacts.

**IxG, IG and GradCAM.**    The methods IxG, IG, and (guided) GradCAM failed completely from the quality perspective in recall when applied to pixel granularity (c.f. Tab. 2). Those are the only ones directly applicable to the image domain, and thus also included the vision encoder in the backward pass. We did not further investigate or fine-tune evaluations to any method. For better comparison we therefore only present the token-based comparisons. All methods are evaluated with the same metrics and therewith give us a reasonable performance comparison without additional customization or configuration.

**Details on Results.**    For a fair comparison, all experiments were executed on a single GPU, as scaling naturally extends all methods. We also want to highlight that we did not optimize the methods for performance further but rather adopted the repositories as they were. The memory inefficiency of gradient-based methods arises from the backward pass. A maximal memory performant representative is the Single-Layer-Attribution method IxG, which only computes partial derivatives on the input with respect to the loss. Even this approach increases the memory requirement beyond an additional $50\%$ and fails for the scaling experiments up to 34B.

In Fig. 5 we ran Chefer with a full backward pass. We adopted this to the minimum amount of gradients (we saw) possible and plot the full scaling benchmark below in Fig 7[3]. The key message remains the same. With the given IxG argument, we do not see much potential in improving memory consumption further.

The methods IxG, IG and GradCam are integrated using the library Captum [15]. We expect them to be implemented as performant as possible. IntegratedGradients is a perturbation method on the input, integrating changes over the gradients. The implementation at hand vastly runs OOM. Finally GradCam is a method specialized on CNN networks and therefore does not work for text only (or varying sequence lengths). It requires the least amount of resources but also produces poor results, without further investigations.

**ATMAN Parallelizability.**    As a final remark, we want to recall again that the runtime measured in the sequential execution can be drastically reduced due to its parallelizability, i.p., as it only requires forward passes. For sequence length 1024, we measured 1024 iterations in order to explain each token. However note that ATMAN can also be applied to only parts or chunks of the sequence (c.f. Sec. A.8), in contrast to gradient methods. Moreover, all tokens to explain can be computed entirely in parallel. In a cluster deployment, these can be distributed amongst all available workers. On top, it can be divided by the available batch size and true pipeline-parallelism (c.f. Fig. 6).

---

[3]Setting *requires_grad=False* to every but the attention tensors.

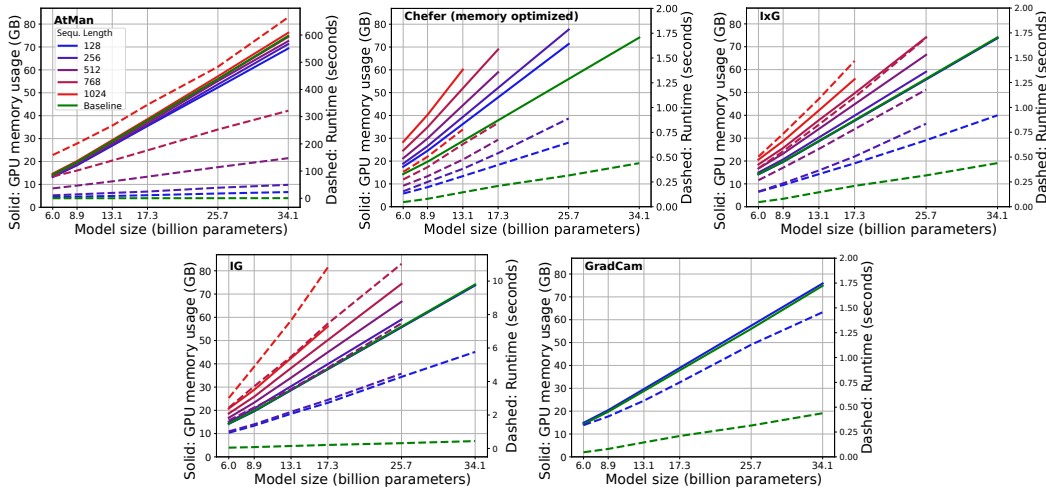

Figure 7: Performance comparison of the explainability methods over various model sizes (x-axis) executed on a single 80GB memory GPU, with fixed batch size 1. Solid lines refer to the GPU memory consumption in GB (left y-axis). Dashed lines refer to the runtime in seconds (right y-axis). Colors indicate experiments on varying input sequence lengths. As baseline (green) a plain forward pass with a sequence length of 1024 is measured. Note that GradCAM can only be applied to the vision domain, it is therefore fixed to 144 tokens. Note that it already consumes as much memory as a forward pass of 1024 tokens. (Best viewed in color.)

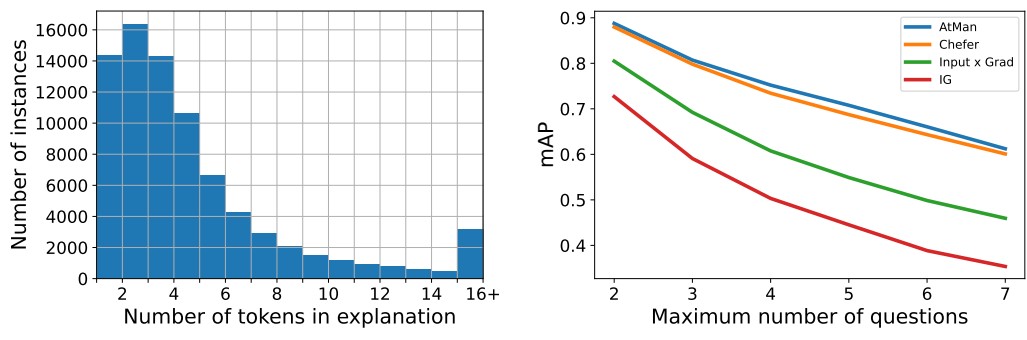

(a) Histogram of explanation token length.

(b) mAP for all methods, grouped by number of question/ answer pairs in the dataset.

Figure 8: Further evaluations on the SQuAD dataset. (Best viewed in color.)

## A.2 Detailed SQuAD Evaluations

This sections gives more detailed statistics on the scores presented in Tab. 1. First Fig. 8a is the histogram of the token lengths of all explanations. Fig. 8b is the mAP score for all methods on the entire dataset, grouped by the number of questions occuring per instance.

## A.3 Detailed OpenImages Evaluations

This section gives more detailed statistics on the scores presented in Tab. 1. Fig. 9 is the histogram of the fraction of label coverage on all images. Fig. 10a and 10b are boxplots for all methods on the entire dataset, for mean average precision as well as recall. Note that the methods IG, IxG and GradCam can be evaluated on image-pixel level, contrary to ATMAN and Chefer. However their performances (i.p. wrt. recall) significantly degenerate, c.f. Tab. 2, Fig. 15.

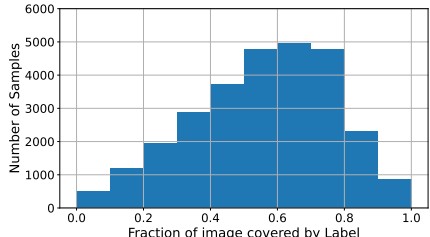

Figure 9: Histogram of percentage of label coverage of the images.

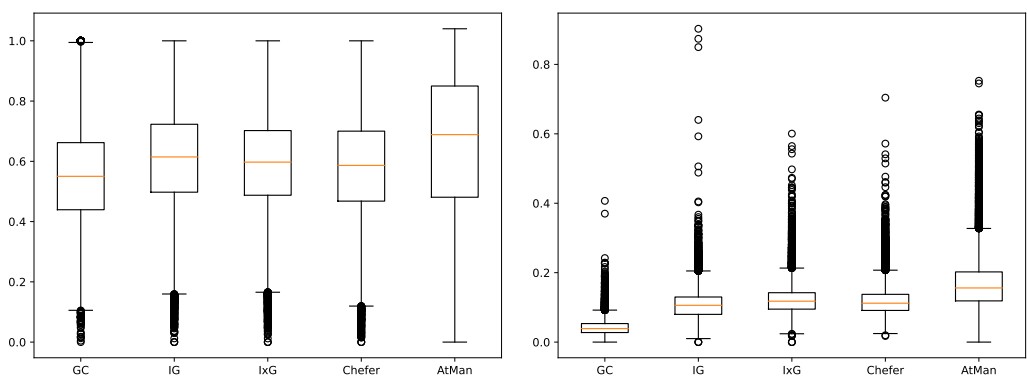

(a) mAP Boxplot for all methods of all images.  (b) mAR Boxplot for all methods of all images.

Figure 10: Further evaluations on the OpenImages dataset. (Best viewed in color.)

## A.4 Discussion of Cosine Embedding Similarity and Parameter values

We fixed the parameter $\kappa = 0.7$ of Eq. 6 and $f = 0.9$ of Eq. 4 throughout this work. They were empirically concluded by running a line sweep on a randomly sampled subset of the OpenImages dataset once, c.f. Fig. 11. Note that pure token-masking ("mask") always leads to worse performance compared to our more subtle concept-suppression operations. We use the same parameters throughout this work, however, parameters may be optimizable depending on the use case and dataset.

In Fig. 12a and 12b we compare the mean average precision and recall scores for OpenImages for both variants, with and without correlated token suppression (to threshold $\kappa$). Clearly the latter outperforms single token suppression.

The following Fig. 13 shows visually the effect on weak image segmentation when correlated suppression of tokens is activated, or when using single token suppression only. Notice how single token only occasionally hits the label, and often marks a token at the edge. This gives us a reason to believe that entropy is accumulated around such edges during layer wise processing. This effect (on these images) completely vanishes with correlated suppression of tokens.

## A.5 Variation Discussion of the method

Note that the results of Eq. 4 are directly passed to a softmax operation. The softmax of a vector $\mathbf{z}$ is defined as
$$\text{softmax}(\mathbf{z})_i = e^{z_i} / \sum_j e^{z_j}.$$

In particular, the entries $z_k = 0$ and $z_l = -\infty$ will yield to the results $\text{softmax}(\mathbf{z})_k = 1/\sum_j e^{z_j}$ and $\text{softmax}(\mathbf{z})_l = 0$. So one might argue as follows: If we intent to suppress the entropy of token $i$, we do not want to multiply it by a factor $0$, but rather subtract $-\infty$ of it. I.e. we propose the modification
$$(H)_{u,*,*} = (H)_{u,*,*} + \log(f). \tag{7}$$

The only problem with this Eq. 7 is, that it skews the cosine neighborhood factors. While we experienced this working more naturally in principle, for hand-crafted factors, we did not get best

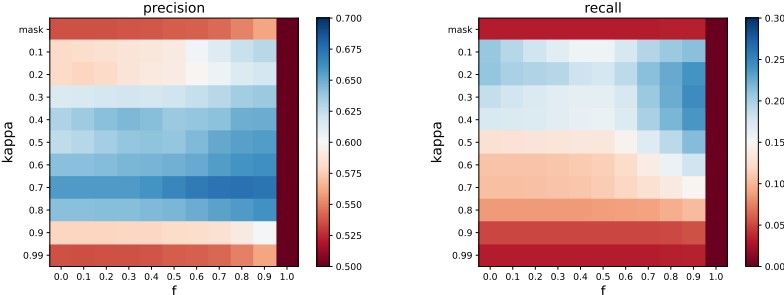

Figure 11: Parameter sweep for Eq. 6. It can be observed that indeed cosine similarity (y-axis) along with more subtle modification (x-axis) outperforms other configurations, i.p. row "mask" that entirely masks single tokens (i.e. multiplies $1 - f$). This result indicates that indeed conceptual entropy is scattered across tokens, that need to be suppressed at once.

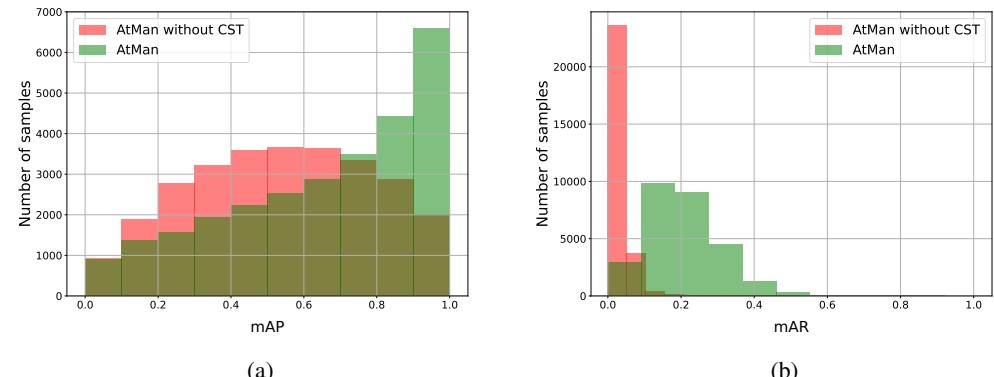

Figure 12: Histogram metric evaluation on OpenImages for ATMAN with (green) and without (orange) correlated suppression of tokens. (Best viewed in color.)

performance in combination with Eq. 6. In the following Fig. 14a and 14b, we show analogous evaluations to Fig. 12a and 12b. It is in particular interesting that the mode without correlated tokens slightly improves, while the one with slightly decreases in scores, for both metrics.

## A.6 Artifacts and failure modes

Merged into A.12 (kept here for reference).

## A.7 Qualitative comparison weak image segmentation

In the following Fig. 15 we give several examples for better comparison between the methods on the task of weak image segmentation. To generate the explanations, we prompt the model with "<Image> This is a picture of " and extract the scores towards the next target tokens as described with Eq. 5 for ATMAN. For multiple target tokens, these results are averaged. In the same fashion, but with an additional backpropagation towards the next target token, we derive the explanations for Chefer and the other gradient methods.

## A.8 Application to document q/a

In Fig. 16 we apply ATMAN on a larger context of around 500 tokens paragraph wise. The Context is first split into chunks by the delimiter tokens of ".", ",", "\n" and " and". Then iteratively each chunk is evaluated by prompting in the fashion "$\{Context\}$ Q:$\{Question\}$ A: " and the cross entropy extracted towards the target tokens, suppressing the entire chunk at once, as described in Sec. 3. It can be observed that the correct paragraphs are highlighted for the given questions and expected targets.

Table 2: OpenImages (OI) evaluations on pixel granularity.

|          | IxG  | IG   | GC   |
|----------|------|------|------|
| mAP      | 38.0 | 46.1 | 56.7 |
| $mAP_{IQ}$ | 34.1 | 45.2 | 60.4 |
| mAR      | 0.2  | 0.3  | 0.1  |
| $mAR_{IQ}$ | 0.1  | 0.1  | 0.1  |

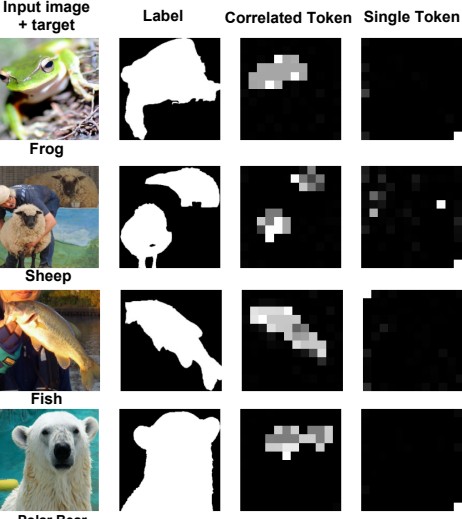

Figure 13: Example images showing the effect of correlated suppression as described in Correlated Token Attention Manipulation Sec. 3.3 and Single Token Attention Manipulation Sec. 3.2. (Best viewed in color.)

In particular, one can observe the models interpretation, like the mapping of formats or of states to countries. Note in particular that it is not fooled by questions not answered by the text (last row).

## A.9    Attention Rollout, Flow, other Perturbation method

Attention Flow constructs a graph representing the flow of attention through the model. Unfortunately this approach is computationally too intense for evaluation on large language models; Chefer et al. (2021) already ran into issues on smaller BERT models. LIME [24] and SHAP [19] are perturbation methods closely related to ATMAN. However they result in several $100.000$ forward trials which is inpractical for use with large language models.

Finally, we ran additional experiments for Attention Rollout [1]. Note that the authors stress its design for encoder classification architectures, while we explore the applicability to generative decoder models in this work. Further, Rollout is by default class-agnostic, in contrast to ATMAN or the studied gradient methods. We adopted Attention Rollout by appending the target to the prompt, and only aggregating attentions over those last target sequence positions. Note that in contrast to the other studied methods, Rollout disregards the output distribution.

We observed large artifacts on the explanations at the positions of the last 2 image patches; apparently most attention aggregation take place at those positions, which may be due to the causal mask. On OpenImages our adoption of Rollout achieves a $mAP$ score of $43.6$ and a $mAR$ of $1.3$ (c.f. AtMan with $65.5$ and $13.7$). Scores do increase when removing the last image patch, however, this also entirely erases explanation on this patch. On Squad we achieved $mAP$ of $23.1$ and a $mAR$ of $53.4$ (AtMan: $73.7$, $93.4$).

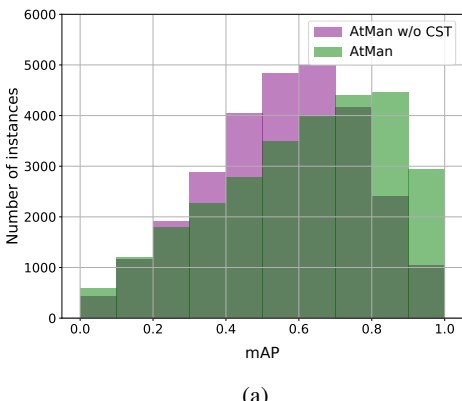 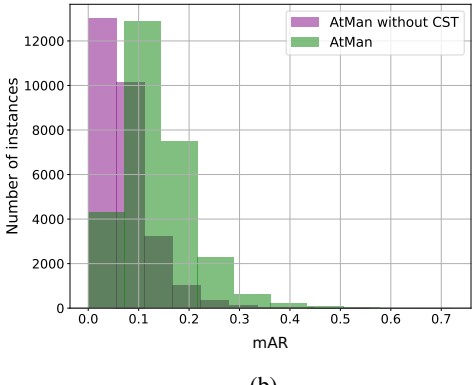

|              |              |
|:---:|:---:|
| (a) | (b) |

Figure 14: Histogram of the evaluation metrics on OpenImages for ATMAN with and without correlated token suppression. (Best viewed in color.)

Table 3: Evaluation of XAI on BLIP model.

|            | Chefer | ATMAN |
|------------|--------|-------|
| mAP        | 59.7   | 64.6  |
| $\text{mAP}_{IQ}$ | 60.3   | 66.1  |
| mAR        | 20.2   | 26.4  |
| $\text{mAR}_{IQ}$ | 19.4   | 28.4  |

## A.10    Demarcation to other architectures, BLIP experiments

To answer the question of *"why we think that AtMan could be generalized to other (generative) multimodal Transformers"*, consider the following arguments.

We conduct for the first time experiments on generative decoder models. The underlying MAGMA model extends the standard decoder transformer architecture, having causal attention masks in place, by additional sequential adapters that could potentially shift distributions. However, the proposed attention manipulation still leads to a robust perturbation method — in text-only as well as image-text modality. We thus conjecture that the skip connections lead to a somewhat consistent entropy flow throughout the network. OFA (< 1B) and BLIP rely on standard encoder-decoder architectures —even without adapters— and the same skip connections. Furthermore, both models process the input image in a similar fashion to obtain "image tokens" forwarded throughout the transformer at the same positions.

BLIP, in contrast to OFA, separates the processing of the image, the prompt, and the answer generation into 3 different models —standard encoders, only the answer generation model being a decoder. The different outputs are cross-attended. We apply ATMAN only on the vision-encoder model.

For comparison, we furthermore evaluate Chefer on BLIP, its attention aggregation being only applied to the vision encoder as well. Tab. 3 compares these two methods, showing that ATMAN still outperforms Chefer. Note, however, that Chefer achieves better scores compared to being applied to the MAGMA model.

On language models, we found that "chat"-finetuned-versions are more applicable to this method, most likely as they are specifically trained to respect the given input. In general, prompt-tuning instructions may improve the results, such as "Answer with respect to the given context".

## A.11    Metrics

All recall and precision metrics in this work are based on the normalized continuous values as follows: For relevancy scores $x$ and binary segmentation label $y$, let $\widetilde{x} = x / \max_i(x_i)$. We compute $AP = (\sum_i x_i y_i) / \sum_i x_i$ for precision and for recall $AR = (\sum_i \widetilde{x}_i y_i) / \sum_i y_i$. The final mean scores

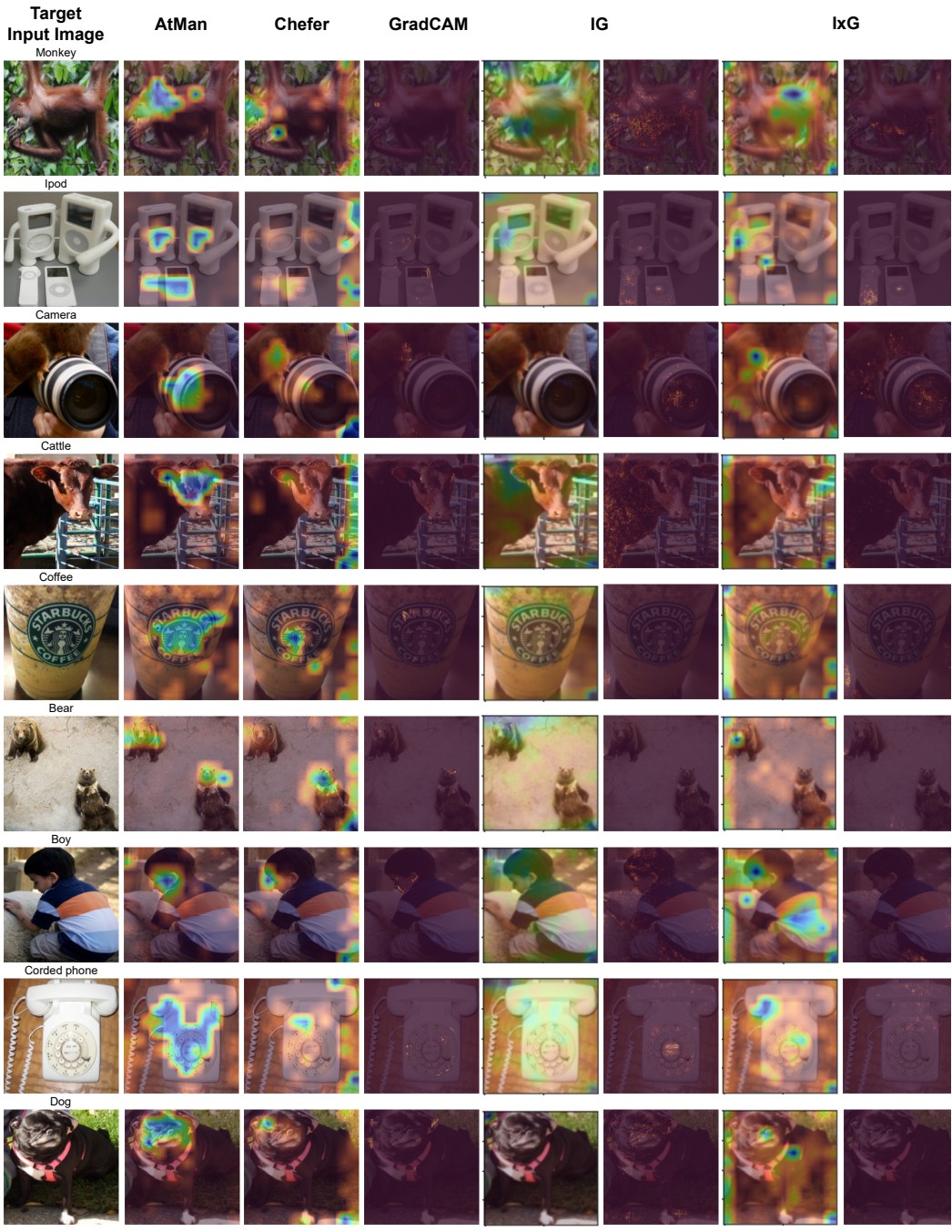

Figure 15: Weak image segmentation comparison of several images for all methods studied in this work. Note that IG and IxG are presented in token (left) and pixel-granularity (right). (Best viewed in color.)

are the average of all samples. This yields a comparable untuned metric also catching the robustness of the methods.

## A.12  Interpretability of "complex facts" and failure modes

All experiments were executed in the generative setting, c.f. Eq. 1,2 and the remark to "global explanations" at the end of Sec. 3. Fig. 1a shows "real text generation" and the accumulation of explanations on the input image during the model's generation process. For comparison benchmarks,

| Question (Q) Target (T) | Context |
|---|---|
| **Q:** What is the pay per hour?

**T:** $10.00 | This contract, dated on the 2nd day of November in the year 1943, is made between TomatoJuicers Corp. and Ronald Smith. This document constitutes an employment agreement between these two parties and is governed by the laws of the state of Michigan. WHEREAS the Employer desires to retain the services of the Employee, and the Employee desires to render such services, these terms and conditions are set forth. IN CONSIDERATION of this mutual understanding, the parties agree to the following terms and conditions: Employment The Employee agrees that he or she will faithfully and to the best of their ability to carry out the duties and responsibilities communicated to them by the Employer. The Employee shall comply with all company policies, rules and procedures at all times. Position As a jury clerk, it is the duty of the Employee to perform all essential job functions and duties. From time to time, the Employer may also add other duties within the reasonable scope of the Employee's work. Compensation As compensation for the services provided, the Employee shall be paid a wage of $10 (per hour) and will be subject to a quarterly performance review. All payments shall be subject to mandatory employment deductions (State & Federal Taxes, Social Security, Medicare). Benefits The Employee has the right to participate in any benefits plans offered by the Employer. The employer currently offers Unlimited PTO, Medical and Dental insurance. Access to these benefits will only be possible after the probationary period has passed. Probationary Period It is understood that the first 2 months of employment constitutes a probationary period. During this time, the Employee is not eligible for paid time off or other benefits. During this time, the Employer also exercises the right to terminate employment at any time without advanced notice. |
| **Q:** What is the date on the contract? Please answer in dd-mm-yy format

**T:** 2-11-43 | This contract, dated on the 2nd day of November in the year 1943, is made between TomatoJuicers Corp. and Ronald Smith. This document constitutes an employment agreement between these two parties and is governed by the laws of the state of Michigan. WHEREAS the Employer desires to retain the services of the Employee, and the Employee desires to render such services, these terms and conditions are set forth. IN CONSIDERATION of this mutual understanding, the parties agree to the following terms and conditions: Employment The Employee agrees that he or she will faithfully and to the best of their ability to carry out the duties and responsibilities communicated to them by the Employer. The Employee shall comply with all company policies, rules and procedures at all times. Position As a jury clerk, it is the duty of the Employee to perform all essential job functions and duties. From time to time, the Employer may also add other duties within the reasonable scope of the Employee's work. Compensation As compensation for the services provided, the Employee shall be paid a wage of $10 (per hour) and will be subject to a quarterly performance review. All payments shall be subject to mandatory employment deductions (State & Federal Taxes, Social Security, Medicare). Benefits The Employee has the right to participate in any benefits plans offered by the Employer. The employer currently offers Unlimited PTO, Medical and Dental insurance. Access to these benefits will only be possible after the probationary period has passed. Probationary Period It is understood that the first 2 months of employment constitutes a probationary period. During this time, the Employee is not eligible for paid time off or other benefits. During this time, the Employer also exercises the right to terminate employment at any time without advanced notice. |
| **Q:** Which country is this contract based upon ?

**T:** United States | This contract, dated on the 2nd day of November in the year 1943, is made between TomatoJuicers Corp. and Ronald Smith. This document constitutes an employment agreement between these two parties and is governed by the laws of the state of Michigan. WHEREAS the Employer desires to retain the services of the Employee, and the Employee desires to render such services, these terms and conditions are set forth. IN CONSIDERATION of this mutual understanding, the parties agree to the following terms and conditions: Employment The Employee agrees that he or she will faithfully and to the best of their ability to carry out the duties and responsibilities communicated to them by the Employer. The Employee shall comply with all company policies, rules and procedures at all times. Position As a jury clerk, it is the duty of the Employee to perform all essential job functions and duties. From time to time, the Employer may also add other duties within the reasonable scope of the Employee's work. Compensation As compensation for the services provided, the Employee shall be paid a wage of $10 (per hour) and will be subject to a quarterly performance review. All payments shall be subject to mandatory employment deductions (State & Federal Taxes, Social Security, Medicare). Benefits The Employee has the right to participate in any benefits plans offered by the Employer. The employer currently offers Unlimited PTO, Medical and Dental insurance. Access to these benefits will only be possible after the probationary period has passed. Probationary Period It is understood that the first 2 months of employment constitutes a probationary period. During this time, the Employee is not eligible for paid time off or other benefits. During this time, the Employer also exercises the right to terminate employment at any time without advanced notice. |
| **Q:** How many wheels does a bike have?

**T:** Two | This contract, dated on the 2nd day of November in the year 1943, is made between TomatoJuicers Corp. and Ronald Smith. This document constitutes an employment agreement between these two parties and is governed by the laws of the state of Michigan. WHEREAS the Employer desires to retain the services of the Employee, and the Employee desires to render such services, these terms and conditions are set forth. IN CONSIDERATION of this mutual understanding, the parties agree to the following terms and conditions: Employment The Employee agrees that he or she will faithfully and to the best of their ability to carry out the duties and responsibilities communicated to them by the Employer. The Employee shall comply with all company policies, rules and procedures at all times. Position As a jury clerk, it is the duty of the Employee to perform all essential job functions and duties. From time to time, the Employer may also add other duties within the reasonable scope of the Employee's work. Compensation As compensation for the services provided, the Employee shall be paid a wage of $10 (per hour) and will be subject to a quarterly performance review. All payments shall be subject to mandatory employment deductions (State & Federal Taxes, Social Security, Medicare). Benefits The Employee has the right to participate in any benefits plans offered by the Employer. The employer currently offers Unlimited PTO, Medical and Dental insurance. Access to these benefits will only be possible after the probationary period has passed. Probationary Period It is understood that the first 2 months of employment constitutes a probationary period. During this time, the Employee is not eligible for paid time off or other benefits. During this time, the Employer also exercises the right to terminate employment at any time without advanced notice. |

Figure 16: Showing ATMAN capabilities to highlight information in a document q/a setting. The model is prompted with "{Context} Q:{$Question$} A: " and asked to extract the answer (target) of the given Explanation. Here, ATMAN is run paragraph wise, as described in text, and correctly highlights the ones containing the information. **All Explanations were split into ~ 50 paragraphs (thus requiring only 50 ATMAN forwad-passes)**. In particular it is shown in row 2 that the model can interpret, i.e. convert date-time formats. Row 3 shows that it can derive from world knowledge that Michigian is in the US. Row 4 shows that the method ATMAN is robust against questions with non-including information. (Best viewed in color.)

Table 4: Performance of scaled MAGMA models.

|        | 6b   | 13b  | 30b  |
|--------|------|------|------|
| VQA    | 60.0 | 62.6 | 64.2 |
| OKVQA  | 37.6 | 38.2 | 43.3 |
| GQA    | 47.4 | 43.7 | 45.6 |

our target was to separate generated explanations from the underlying model's interpretation, by retrieving "obvious facts". Fig. 4 demonstrates qualitatively target-class awareness.

More complex results are shown in Fig. 17. Fig. 17a gives an example where the model is VQA prompted and ATMAN explains both modalities at the same time for the completion token "white". It highlights the words "tub" and "color" of the text, and an edge of the tub in the picture as significant to produce the completion. Note that we find these "joint-modal explanations" to work best when the target token is not just a yes/no answer, c.f. Fig. a'). Further note how these artifacts remain amongst explainability methods such as Chefer, which may be due to the underlying models architecture. We leave these investigations for further research.

In Fig. 17b ATMAN highlights multiple heterogenous concepts to answer the counting task. Note that here the explanation for the completion "two", which actually highlights visually the animals, is found at the token position "animals", and not at the position of the predicted token. ATMAN returns at the same time explanations to all previous context positions, without additional forward passes (in contrast to gradient-based methods).

Finally, we ran evaluations on visual reasoning using the GQA dataset 17. Fig. 17c shows an instance of GQA, in which the model is prompted to answer "Does the man ride a horse?", while the man in the picture actually rides a bike. ATMAN highlights at two different sequence positions of the prompt the respective concepts in the image required to answer the question, once the man, and once the bike. Fig. 17d.1 and d.2 are the resulting IG explanations when applied to explain the same token positions; somewhat it captures parts of the concept but focuses on noise to the edge. Note that many samples of this dataset are rather difficult to answer, it is often not clear on what token position an explanation is supposed to be found. In particular, often ambiguous concepts occur multiple times in the image and are further restricted to the correct instances much later in the text description. This is, in particular, difficult to grasp for generative decoder models due to the causal masking. As the dataset contains annotations for areas as in example Fig. 17c, we ran preliminary quantitative results on 10k randomly chosen instances. ATMAN matches bounding boxes with $mAP$ 52.4 and $mAR$ 24.3. For IG we obtained $mAP$ 33.3 and $mAR$ 27.0.

### A.13  Partial attention manipulation

We apply the same manipulation in each layer. We hypothesised —and first experiments indicate— that one needs a "critical mass", to remove the entire concept entropy from the generation process. This is i.p. due to the skip connections. Only applying it to a single (arbitrary) layer did not yield comparable results. However we did not finalize a conclusion yet and leave this to future work. C.f. A.15, Fig. 18.

### A.14  MAGMA scaling

MAGMA Model performances on standard text-image evaluation benchmarks are shown in Tab. 4.

### A.15  Variants of ATMAN and final remarks

There are many promising variants of ATMAN not yet fully understood. In particular finding the relevant layers and other formulae to manipulate reliably the activations, or to normalize the output scores, could yield further insights into the transformer architecture. We showed how conceptual suppression improves overall performance. For pure NLP tasks, utilizing an embedding model such as BERT to compute similarities could be beneficial. Moreover, it should be possible to aggregate tokens and increase the level of details as needed, to save additional computational time.

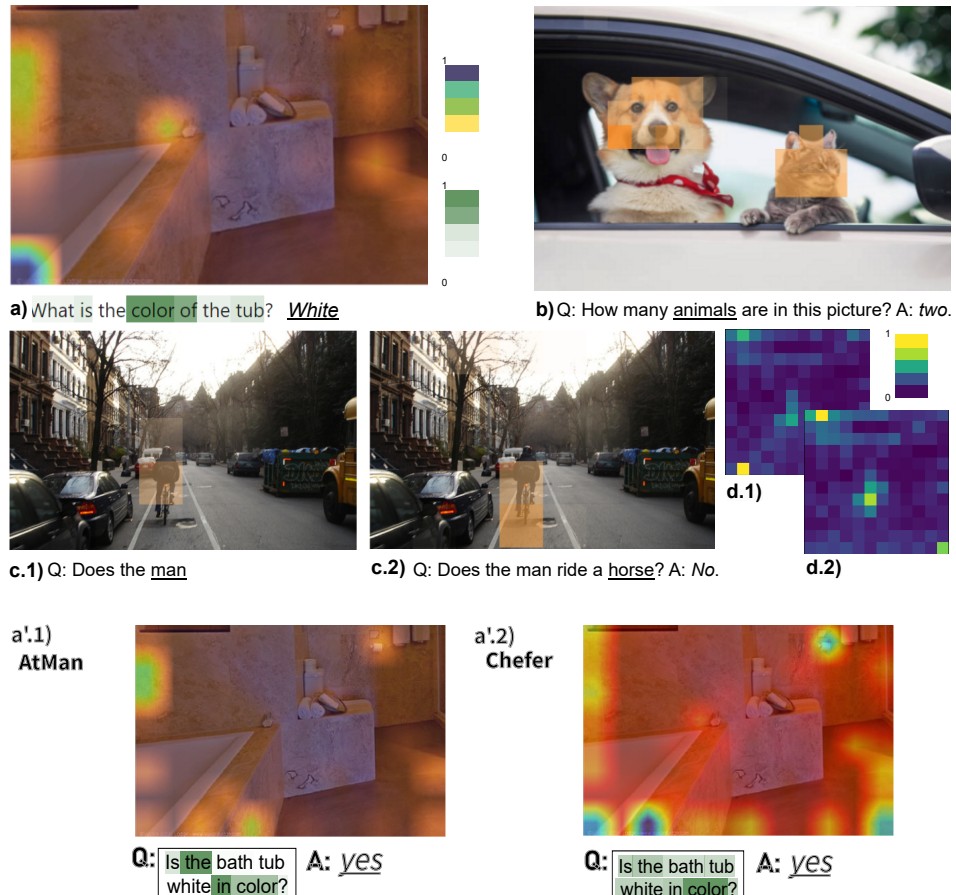

**a)** What is the color of the tub? *White*

**b)** Q: How many animals are in this picture? A: *two*.

**c.1)** Q: Does the man

**c.2)** Q: Does the man ride a horse? A: *No.*

**d.1)**

**d.2)**

a'.1)
**AtMan**

**Q:** Is the bath tub white in color? **A:** *yes*

a'.2)
**Chefer**

**Q:** Is the bath tub white in color? **A:** *yes*

Figure 17: More complex examples showing generative multi-modal explanations (c.f. Sec. A.12 for detailed description). *italic*: model completion; underline: token position for explanation. (Best viewed in color.)

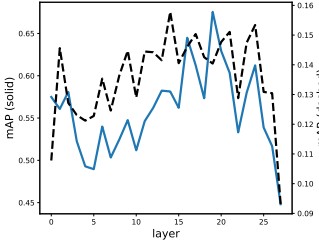

Figure 18: Measured mAP and mAR when applying AtMan only to a single layer, with similarity of the first embedding layer. Evaluated on a subset of openimages. Apparently, some layers contribute more to the explanation than others. "Full AtMan" as proposed yet outperforms on both metrics with scores mAP = 0.66 and mAR = 0.24.

Fig. 18 shows scores when applying ATMAN only to a single layer. I.p. the second half of layers seem more influential to the ATMAN scores, which is in accordance with recent research studying that higher-level concepts form later throughout the network.

In Fig. 19 we applied softmax on the embeddings (still only of the first layer) instead of the proposed thresholded cosine similarity Eq. 6. The proposed cosine similarity clearly outperforms its softmax variant in quality, however visible correlations can be found in the softmax-image as well.

As a final remark, we want to highlight again the rather undefined question of "what an explanation is". Afterall, we show what input leads to a diverging generation of the model faithfully, even if it would not match the expected outcome.

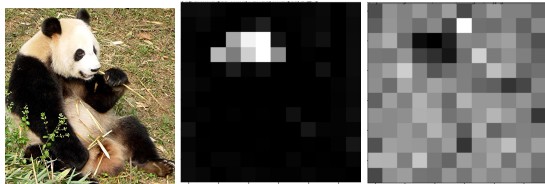

Figure 19: Qualitative example for an explanation of AtMan as proposed (middle) and AtMan when applying the softmax on the embedding instead of cosine similarity (right), as suggested by a reviewer. While some correlation can be found, it is much more noisy.

