# OpenReview forum: "ATMAN: Understanding Transformer Predictions Through Memory Efficient Attention Manipulation"
_NeurIPS.cc/2023/Conference — NeurIPS 2023 poster_

### Official Review · Reviewer_wnDz · 2023-07-04

**Soundness:** 3 good
**Presentation:** 4 excellent
**Contribution:** 2 fair
**Rating:** 5
**Confidence:** 4

**Summary:**

The paper introduces AtMan, a method that manipulates the attention mechanism of generative transformer models, to provide explanations. More specifically, it perturbs raw attention scores instead of the token inputs. The authors provide extensive experiments for different modalities to back their claims regarding better results to established baselines.

**Strengths:**

1. The paper introduces a new approach towards Explainable AI, that does not require calculation of gradients through the network, thus enabling evaluation of larger models with the same hardware.
2. The idea of manipulating attention scores is novel and interesting.
3. The authors embark on an extensive evaluation of their method on different modalities. The universality of the Transformer architecture, enabled them to seamlessly switch between modalities. The methods performs well on a range of architectures (encoder, decoder).

**Weaknesses:**

1. One of the main motivation of this work is to enable more widespread adoption of explainable AI methods, via reducing required computational resources. The issue of memory is discussed extensively. Inference speed is only mildly explored, in Figure 5. For large sequence lengths, the method requires upwards of 2 orders of magnitude more time to make the final predictions.
2. The method requires one to set new hyperpatameres, namely $f$ and $\kappa$. The effect of those with respect to e.g. sequence length or model architecture, are not discussed. For example, different sequence lengths may lead to different softmax entropy and thus different $f$ values might be required.
3. According to Eq. (4), tokens are masked by setting the corresponding $H$ values to 0. This does by no means lead to masking of the tokens, as briefly explained in Appendix A5. Setting the values to the value $0$ constitutes another hyperparameter, that needs to be set for the proposed method. In principle any real value could have been chosen (In Appendix A5 the authors explore the values $0$ and $-\infty$). The range of the possible values was not explored at all.
4. In general a lot of approximations are made for the derivations of the last method, with possible alternatives not discussed. Why was Eq. (4) chosen to mask tokens? Why is the cosine similarity of the embeddings a good measure to suppress common tokens? In [1] for example, the cosine similarity between keys is used as a more suitable measure. The cosine similarity is calculated based on the embeddings of what layer? Is there a benefit from calculating this similarity on different layers? There is a lot of work on what type of information token embeddings carry for different layers in a Vision Transformer.

[1] Bolya, Daniel, et al. "Token merging: Your vit but faster." arXiv preprint arXiv:2210.09461 (2022).

**Questions:**

1. Line 62 states that the source code is released although I could not find it in the supplementary.
2. Line 33-34 states that memory consumption leads to uneconomical productive deployment. This is a overstatement, especially given the computational requirements of the new proposed method.
3. A lot of literature has been exploring the effect of attention scores in the context of Stable Diffusion, e.g. [2] and follow-ups.
4. Is there a minus missing before $L^{\text{target}}$ in Eq. (2)?
5. Line 245, it is not clear to me how the mean of generated scores affects the final predictions. Are some tokens in the target label more important than others as expected?
6. Is there a way to prioritize checking some tokens first instead of exhaustively trying out all tokens to find the ones that explain better the target? This might help to reduce the inference speed. You can think about having a method to rank tokens to check, and then create a plot of computational budget vs performance, where you check iteratively more tokens. You could also think of doing this in a more hierarchical way, similar to what is proposed in Appendix A8.

[2] Hertz, Amir, et al. "Prompt-to-prompt image editing with cross attention control." arXiv preprint arXiv:2208.01626 (2022).

**Limitations:**

The paper proposes a way to manipulate attention scores to explain prediction of Transformer models. Manipulating these scores is done a token at a time, leading to inference speed that scales with the sequence length, and is orders of magnitude bigger than previous approaches. Although a lot of experiments are presented, some of the design choices are not adequately justified.

---

> ### Author Rebuttal · Authors · 2023-08-09
>
> Dear reviewer,
>
> thank you for the invested time reviewing our work and your valuable feedback.
>
> W1 (inference speed)
>
> As discussed in the global review, we want to highlight once again, that this is the first XAI method deployable at all for large transformer networks, without additional allocation on deployment resources. For instance, models exceeding 9B parameters are not deployable on a single 80GB GPU using current methods such as Chefer et al. To emphasize this more throughout our paper and prevent further misunderstanding we updated the main text and Figure 5. Please see attached Fig 5 in the supplemented pdf. Furthermore, we now illustrate the computation time using AtMan and previous methods on a simple deployment layout, see Fig 16. To explain an image, it can be observed that AtMan takes about 1 magnitude longer in total processing time, but can still be processed in seconds of total compute with pipe-parallel 4 and batch-size 16. Note that the forward passes are entirely independent, and thus the processing time can further be pushed to subseconds, with more idle workers. When chunking input to paragraphs as described in Appendix section A8, Figure 14, moderately larger prompts with around 500 tokens and 50 ‘explanation passes’ remain around 10 computation seconds (that can be executed in parallel).
>
> W2 (factors $f$ and $\kappa$)
>
> Thank you for the good question. Indeed $f$ and $\kappa$ depend on model architecture and the type of dataset. To determine these factors we ran the following search on openimages, cf attached Fig. 17. As it can be observed, $f=0.9$ and $\kappa=0.7$ leads to the best values in mean precision and recall. We used the same factors for the tasks VQA and GQA (A12 Fig15), even though the style of prompts largely varies. We will add a discussion on this limitation and how to find fitting parameters to the main text.
>
> Regarding sequence length, computing the precision and recall as described in A11 is invariant against a shift in e.g. magnitude of the actual values of the influence function --- the explanation becomes the (normalized) heatmap.
> We gave qualitative examples on large prompt lengths (500+ tokens) with A8 Fig14. As it can be observed, along with (varying sequence lengths) in the Squad benchmark, the results remain stable. We will clarify this further in the submission by extending Fig 17 to BLIP and Squad.
>
> W3 (masking tokens)
>
> Thank you for this valuable feedback. However, there seems to be a misconception here. We never mask out tokens entirely by setting them to 0 (or -$\infty$). To provide further clarifications for the readers, we additionally show that masking tokens entirely would even yield degenerated performance as shown in the top row ‘mask’ of the Fig 17 (c.f. W2).
>
> W4 (questions regarding Eq.4)
>
> Thank you for this remark. We will add further discussions on approximations and alternatives to the main paper. In particular, we will be more precise regarding your following questions:
>
> a) Why Eq. (4)?
>
>    It is the most generic equation possible, i.p. as we want to apply cosine similarity afterward. The function of ‘the causal mask’ in fact is ‘to apply suppression’ already, such that attention cannot attend to ‘future tokens in the sequence’. It is therefore a reasonable position to apply perturbation.
>
> b) (cosine similarity)
>
>    We compute the cosine similarity directly after the embedding layer. The vision encoder of MAGMA is a pre-trained CLIP-like model (c.f. line 133) that is trained on a contrastive loss and thus expected to already produce comparable embeddings (and our results reinforce). They are consistently applied throughout the network.
>    Improving the similarity matching we proposed is indeed an interesting line of research. We will add this to the discussion section. Note that we study decoder models contrary to encoders like ViT. The applied causal mask most likely destroys the similarity of ‘future’ tokens required, when calculating similarity in middle layers. Initial experiments did not succeed, but will be discussed.
>
> Q1 (source code)
>
> Indeed, we just sent the code to the AC, we hope they will forward them to you soon.
>
> Q2 (runtime)
>
> In contrary to previous gradient-based methods, AtMan is indeed practically performant. Please cf. W1.
>
> Q3 (stable diffusion)
>
> Thank you for the references, we will add Hertz, Amir, et al. and further references to a dedicated related work section on the effect of manipulating attention scores.
>
> Q4 (Eq 2)
>
> Mathematically the sign is not necessary, however, we will include it for the sake of readability.
>
> Q5 (mean approximation)
>
> We used the mean as we assumed for simplification standard Gaussian distribution and tried to model the expectation value. Indeed we will further investigate and discuss it. Due to the generative token-wise process, surely the first token of each ‘concept word’ carries the most entropy.
>
> Q6 (token reduction)
>
> Indeed ways to prioritize tokens could lead to performance improvements. However, we would like to emphasize again that AtMan’s inference speed is not impractical, c.f. W1.
> Token aggregation is indeed possible, c.f. appendix section A8. Exploring this further would be interesting follow-up work. In particular, exploiting symmetric cosine similarity or adopting ‘token merging’ seems promising.
>
> Limitation
>
> While we agree that the current implementation of AtMan leaves improvements for inference speed, we would like to highlight again that inference speed is not orders of magnitude bigger than previous approaches. In fact, AtMan is the first deployable XAI method not requiring further infrastructure costs, as it only relies on normal forward-pass workers cf. global review.
> We will clarify the discussion in the paper and in particular add Fig 16 showing practical real-time performance of average prompts.
> The discussed design choices and parameter justifications will be added to the paper.
>
> Thank you again,
> the authors.

---

> > ### Comment · Reviewer_wnDz · 2023-08-14
> >
> > Thank you for the additional details and the clarity in the text, these were really helpful.
> >
> > Figure 17 adds to the understanding of the hyperparameters and I think would help to be discussed in the paper itself.
> >
> > Further limitations are further discussed by the authors, namely 1. Inference speed and 2. Numerous simplifications made, guided namely by preliminary empirical results. Regarding (1) authors provide Figure 16 which I find particularly misleading, given that multiple workers and   batch-decoding is applied only for their method. Regarding (2) there are numerous choices made are not always intuitive or sufficiently explained.
> >
> > I do however find the merit in the approach and believe the new results have demonstrated some potential for applicability and some benefits compared to baselines. I have updated by score.

---

> > > ### Author Response · Authors · 2023-08-21
> > >
> > > Thank you again,
> > > we indeed will add a more sophisticated discussion of Fig 17 to the paper itself.
> > >
> > > W.r.t. (1), Fig 16: Note that (pipe-)parallel execution on gradient based methods is not applicable at all, as it _multiplies_ the already exceeding memory consumption ---the primary measurement criterion for this study--- and further requires a 'warm-up' phase filling the execution pipeline, to further reduce the computation time. The latter is usually not fully filled by practical model queries.
> > > The same technique, however, is inherently applied to plain-forward passes. It as such compares 'practical query execution times' as it was requested, and further demonstrates the computability of the proposed method (at all).
> > > Other parallel execution schemes like model-parallelism, on the other hand, would improve both methods equally. We will further clarify the caption.
> > >
> > > W.r.t. (2), sorry for not being able to fully meet your expectations. Some parts indeed are directed by empirical assessments and provide scope for subsequent research.
> > > Your review already helped us rendering the paper more complete. Please let us know about any concrete aspects we could improve our work further with.
> > >
> > > all the best
> > > the authors

---

### Official Review · Reviewer_FKKY · 2023-07-06

**Soundness:** 3 good
**Presentation:** 3 good
**Contribution:** 2 fair
**Rating:** 6
**Confidence:** 3

**Summary:**

The authors proposed a new resource-friendly modality-agnostic perturbation method named AtMan to explain the predictions of generative transformers. AtMan suppresses or amplifies the attention entry relying on the cosine similarity neighborhood to evaluate the influence of each token, and it saves many memory resources compared with current gradient-based methods, which means that it's more suitable for large model inference developments. Empirical results show that AtMan can do language/visual reasoning better than the current state-of-the-art gradient-based methods.

**Strengths:**

1. AtMan is simple but useful, and the idea that changing perturbation space from the raw input to the embedding space is important. It can perform on decoder as well as encoder architectures and the empirical results are supportive.
2. The core contribution of AtMan is that it reduce the memory cost and can be applied to the large models, which will be important for the future works of explainability of large models. Now it can be applied to 30B multi-modal model but previous methods fail.
3. The paper is well written and easy to follow.

**Weaknesses:**

1. The paper mentions that the $\kappa$ parameter in AtMan is set to be $0.7$, but it may be important for the performance of AtMan, so it is better to do additional experiments to show the effect of different kappa on the experimental results on different datasets. Furthermore, the choice of (6) is not unique, there are some other ways like using the softmax value of $s_{i,k}$ to be $f_{k,*}^i$ and using temperature as a hyper-parameter. It's better to compare these ways.
2. The increase in the time cost of AtMan is significant compared with gradient-based methods, so it's better to show the time-cost figure after pipeline-parallel execution.
3. The results of the locality are important, maybe the authors can try to add them (like figure 10 in Appendix) into the main paper.

**Questions:**

Similar to Weakness-1 and Weakness-2.


**Limitations:**

The author mentions that their work provides reference methods for future work, including explanatory studies of current generative models impacting our society. The author doesn't mention other potential negative societal impacts.

---

> ### Author Rebuttal · Authors · 2023-08-09
>
> Dear reviewer,
>
> thank you for your valuable and very inspiring feedback.
>
>
> W1 (parameters $f$ and $\kappa$)
>
> We apologize for not showing that figure in the appendix already. Please find attached Fig. 17 for the influence of $f$ and $\kappa$ on the openimages benchmark. Note that indeed utilizing the cosine similarity improves upon ‘plain masking of single tokens’, as shown in the ‘mask’ row. We can further run and attach this ablation on the GQA and Squad dataset and BLIP outside the rebuttal period.
> Thank you for the suggestion of iterating on Eq 6. It is indeed a very interesting idea you propose, however, it is not a subtle change. In Fig 2b. we show how the generation process is not degenerated, but rather ‘steered’. The range of the softmax values is usually entirely off and rather incomparable among different samples, as it is normalized to sum up to one. It, therefore, is hard to fix a single temperature to a large benchmark dataset. We tried some samples that however did not work as well (attached Fig 19). Normalizing the softmax by dividing by its max value is almost like returning to the original cosine similarity, so we are not sure how to proceed here.
> We want to foster exactly those discussions by making the source code publicly available and inviting collaborations.
>
> W2 (inference speed)
>
> Thank you for this remark, in the main paper we tried to do the fairest comparison possible, as this plot highly depends on the actually available resources. Nevertheless, we run through reasonable assumptions on those and added exactly this point with attached Fig 16; showing that indeed this algorithm is very well computable and not that far off in magnitudes.
>
> W3 (locality)
>
> Thank you for this suggestion. We agree and will add it to the main paper.
>
>
> Limitation)
>
> We derive an XAI method correlating input to output tokens for the otherwise black-box generations. These first levels of explanations pave the way for exploring e.g. underlying trained, potentially harming, biases when using machine-learned processes.
>
> The underlying process of AtMan includes suppression (or facilitation) of concepts found in the input, to steer the models generation. Indeed the steerable nature may as well be used to also correct those findings, or be abused to increase them. We agree that this should be further discussed as potential negative societal impact and will add this concern to the discussion.
>
> Again thank you for your valuable feedback,
> the authors

---

> > ### Comment · Reviewer_FKKY · 2023-08-17
> >
> > Thank you for your feedback. I'm glad to see that Figures 16 and 17 effectively address my concerns, highlighting an acceptable time-cost trade-off and the reason for choosing the value of kappa. I suggest the authors include these in the revised paper. However, I believe the caption of Figure 16 could be made more understandable by providing clearer descriptions. Besides, the exploration of the token attention score suppression/facilitation discussion is indeed intriguing. I will maintain my positive score on your work and encourage the author to test other relevant algorithms in the following works.

---

> > > ### Author Response · Authors · 2023-08-21
> > >
> > > Thank you again for your review, we will add the discussions to the paper.
> > > We will further clarify, i.p. given more space, the caption of Fig16.
> > >
> > > Kindly find a final remark to Fig 16 as follows:
> > >
> > > Note that pipe-parallel execution on gradient based methods is not applicable at all, as it multiplies the already exceeding memory consumption ---the criterion in focus of this study--- and further requires a 'warm-up' phase filling the execution pipeline, to further reduce the overall computation time. The latter is usually not fully filled by practical inference queries. The same technique, however, is inherently applied to plain-forward passes, not affected by these limitations. It, as such, compares 'practical query execution times' as it was requested, and further demonstrates the computability of the proposed method (at all). Other parallel execution schemes like model-parallelism, on the other hand, would improve both methods equally.
> > >
> > > all the best,
> > >
> > > the authors

---

### Official Review · Reviewer_sfjw · 2023-07-07

**Soundness:** 3 good
**Presentation:** 2 fair
**Contribution:** 3 good
**Rating:** 6
**Confidence:** 4

**Summary:**

This paper proposed ATMAN, a modality and architecture-agnostic perturbation-based XAI method for generative transformer models. Specifically, ATMAN evaluates the sensitivity of model prediction w.r.t. the perturbation of attention scores on a set of cosine-distance neighboring tokens. Empirical results suggest that ATMAN can effectively identify the relevant input tokens for target prediction, and outperforms the baseline methods in terms of precision and recall.

**Strengths:**

As far as I know, this paper is the first work that identifies the crucial tokens/pixels by attention perturbation. The idea is novel and looks interesting to me. The authors further apply the cosine similarity to collect the related neighboring tokens and reduce the searching redundancy. It helps us to understand how transformer models process input signals and identify crucial ones.

**Weaknesses:**

1. As perturbing every token to evaluate their sensitivity, ATMAN can incur high time complexity to generate the explanation for transformer models. As suggested by Figure 5, the runtime is much higher than the baseline. Therefore, to reduce the time complexity, would there be any solution to evaluate the token/pixel sensitivity in a higher granularity, like dividing the tokens/pixels into clusters?
2. The paper mentioned steering model prediction by applying the attention scores of identified tokens but I did not find any empirical results to claim the advantages of this approach. Could you provide more details and results or correct me if I am wrong?

**Questions:**

Can you provide more model details about Section 3.1? Which black-box model do you evaluate?

**Limitations:**

The limitation is the high runtime cost when evaluating every input token/pixel. It would be great if the authors can extend the analysis into higher granularity.

---

> ### Author Rebuttal · Authors · 2023-08-10
>
> Dear reviewer,
>
> thank you for your valuable feedback and the invested time reviewing our work.
>
> W1 (inference time)
>
> As discussed in the global review, we want to highlight once again, that this is the first XAI method deployable at all for large transformer networks, without additional allocation on deployment resources. For instance, models exceeding 9B parameters are not deployable on a single 80GB GPU using current methods such as Chefer et al. To emphasize this more throughout our paper and prevent further misunderstanding we updated the main text and Figure 5. Please see attached Fig 5 in the supplemented pdf. Furthermore, we now illustrate the computation time using AtMan and previous methods on a simple deployment layout, see Fig 16. To explain an image, it can be observed that AtMan takes about 1 magnitude longer in total processing time, but can still be processed in seconds of total compute with pipe-parallel 4 and batch-size 16. Note that the forward passes are entirely independent, and thus the processing time can further be pushed to subseconds, with more idle workers. When chunking input to paragraphs as described with Appendix section A8, Figure 14, moderately larger prompts with around 500 tokens and 50 ‘explanation passes’ remain around 10 computation seconds (that can be executed in parallel).
> Further note that our perturbation is executed in the transformers hidden space. In generative models such as the here studied MAGMA or BLIP, is already clustered into ‘tokens’ (c.f. line 107). In case of MAGMA, 144 ‘image tokens’ are perturbed, in contrast to the large pixel space. Further aggregation such as adopting ‘token merging’ or exploiting the symmetry of cosine similarity should be possible. We will clarify this further in the main paper.
>
> W2 (steering)
>
> It is unclear what comparison in advantages you’d like to see. Throughout the paper, we always apply the attention score modification to ‘steer’ the generative process and derive the XAI method from the distribution shift in token generations. Table 1 empirically shows how we outperform other state-of-the-art methods in performance.
>
> If you meant a comparison to other available perturbation methods such as SHAP, perturbing directly on the input space is resulting in an uncomputable number of trials (c.f. line 540).
> If you meant a comparison to entirely masking out tokens with AtMan (setting $f=1$), we’d like to point to Fig 10 and the attached Fig 17. The top row ‘mask’ refers to ‘entirely masking’ single tokens, throughout the network, not applying the cosine similarity. It can clearly be observed that simply masking single tokens leads to degenerative performance. We assume this arises from the destructive influence on the trained input structure. Our method on the other hand applies a more subtle manipulation that aims to downgrade ‘a concept entirely’ found in the input. We will include this discussion in the paper. We hope this addressed your concern. Let us know if open questions remain.
>
> Question
>
> We do not clearly understand what you are referring to with ‘black-box’ in section 3.1.
>
> If you meant section 4.1, we indeed missed to clearly describe the model under investigation. Thank you for pointing this out. That would be plain gpt-j, the underlying opensource model MAGMA was trained upon. We will update the manuscript accordingly.
>
> If however you really meant section 3.1, assumably you refer to the title of the cited paper of ‘understanding black-box predictions via influence functions’. For us, there is no black-box (other than the garbled hard-to-trace computations executed). The model (MAGMA) is, in fact, white-box, which we exploit as follows:
> We use Influence functions to give a  measurement based on robust statistics for our derived XAI method, which in fact is the visual heatmap of 'a point’s influence'. We regard transformers to be distribution estimators and aligning with the work of [13], define an influence function (Eq 3), that approximates the input-perturbations influence into changes of the model parameter space, ‘as if the model would not remember the to-probe concept’, and trace the distribution loss change. Here we raised the somewhat natural hypothesis, that the query and key scores are the relevant parameters in observing or altering those input changes of ‘point masses’. Our exhaustive empirical evaluation indeed reinforces this argument. We will clarify this in the main paper.
>
>
>
> Thank you for your very valuable feedback that contributed to improving this submission.
> The authors.

---

### Official Review · Reviewer_YuQG · 2023-07-08

**Soundness:** 3 good
**Presentation:** 4 excellent
**Contribution:** 3 good
**Rating:** 7
**Confidence:** 2

**Summary:**

This paper proposed a new permutation method for Transformer explanation, named ATMAN, which directly manipulates attention scores in the hidden layers of the Transformer model rather than permuting the model inputs. Importantly, the authors established a bridge between ATMAN and the relevance between inputs and output predictions. Experimental results demonstrate the effectiveness and efficiency of ATMAN on interpreting Transformer model on different types of data and different model scales.

**Strengths:**

The proposed ATMAN method is novel, and the experimental results shows that ATMAN significantly improve the effectiveness and efficiency over existing perturbation and propagation methods on interpreting Transformer model on different types of data and different model scales.

**Weaknesses:**

NA.

**Questions:**

NA.

**Limitations:**

NA.

---

> ### Author Rebuttal · Authors · 2023-08-09
>
> Dear reviewer,
>
> we want to express our gratitude for taking the time to review our paper and the appreciation of our work.
> If you have further questions let us know.
>
> Thank you,
> the authors.

---

### Official Review · Reviewer_HHf8 · 2023-07-13

**Soundness:** 2 fair
**Presentation:** 2 fair
**Contribution:** 2 fair
**Rating:** 3
**Confidence:** 4

**Summary:**

This paper proposes atman, a method for understanding the predictions of transformers by masking out the tokenized inputs. The paper evaluates atman on explainability tasks in text pipelines and vision pipelines. The authors argue that atman is superior to gradient-based explanation methods since it uses less memory.

**Strengths:**

The paper is well-written and the contributions are easily understood. The method is simple to implement and understand, and does take less memory footprint than gradient-based methods. The evaluation on explainability benchmarks seem to show that it behaves at par with gradient-based methods. The figures make the method very easy to understand.

**Weaknesses:**

The novelty of the method seems questionable, and it doesn't seem necessary to create an entire "method" with a name to describe the observation that masking out input tokens can be useful in tracing which parts of the input sequence are important. This type of analysis is standard in analyzing the outputs of Transformers, e.g. it is a standard technique in Olsson et al 2022 (https://arxiv.org/abs/2209.11895). To claim that it is a novel method is a bit much in terms of framing.

There may be some novelty in the vision domain in looking at the correlated outputs instead of masking individual patches of the input image, but as far as I can tell this methodological change is not ablated.

These novelty questions point to a weakness in the evaluation and the discussion of related work. I don't have any problem with a simple method (in fact I like it) - but if the method is a first standard trick that most ML engineers would use to understand what is happening when a Transformer makes predictions, then it suggests a different framing and evaluation.

For example: What other "surprisingly simple" interventions can be used to explain predictions? Can you look at embeddings of tokens in the final output? What are the differences (in NLP) between generative-models (BERT) and encoder-only models (GPT)? How are the explanations for vision affected when the correlations are not taken into account? What happens if this intervention is done partway through the backbone instead of at the beginning?

**Questions:**

Suggested list of questions to improve the work:
1. What other "surprisingly simple" interventions can be used to explain predictions?
2. Can you look at embeddings of tokens in the final output?
3. What are the differences (in NLP) between generative-models (BERT) and encoder-only models (GPT)?
4. How are the explanations for vision affected when the correlations are not taken into account?
5. What happens if this intervention is done partway through the backbone instead of at the beginning?

---

> ### Author Rebuttal · Authors · 2023-08-09
>
> Dear reviewer,
>
> thank you for the feedback, however, we kindly disagree with your novelty assessment of our proposed XAI method. First and foremost, our work studies, and compares to, state-of-the-art interpretability research in the field of transformers and neural networks, correlating input token importance to the generated output tokens of the otherwise black-box transformer model. While it is true that masking out input tokens has been explored in the context of transformer analysis to some extent, we do not merely mask tokens. We study ---for the first time--- the adjusted generative behavior of decoder models when amplifying or attenuating (ref Fig 2b) the query and key scores consequently throughout all transformer layers (not only in a single layer). We moreover present large benchmark evaluations for explanations on two different modalities, outperforming current gradient-based state-of-the-art methods. Our contribution lies in providing a structured and systematic approach to leverage this observation for solving a specific problem (XAI), which we believe is novel. Specifically, we introduce attention manipulation as a perturbation technique for XAI and derive it as a form of influence functions and carefully devise its efficient application to autoregressive transformer models.
>
> Fig 10 of the appendix section and the attached Fig.17 (‘mask’ row) demonstrate quantitatively, that “simple” interventions would lead to degenerated performance when comparing changes in the logit distribution as proposed in the submission. Note that we already gave qualitative and quantitive evidence to this with Fig. 3a of the main paper and Fig.10 and 11 in the appendix. We have not come across any previous work that specifically addresses the exact problem and approach described in our submission. Therefore, we would appreciate references to relevant publications, even to ‘best ML engineer practices’. Please also consider the global response to this rebuttal.
>
> We addressed the issue rendering state-of-the-art perturbation methods impractical, perturbing in a large discrete token-space, by shifting this sto a continuous factor value space.
>
> With regard to the reference of Olsson et al, the induction-heads proposal indeed is an interesting line of work (even though it seems unpublished to this date). Note that we do not claim to have an explanation for the behavior or responsibilities of generic architecture components, in particular, their development during training. We empirically show and evaluate rigorous consistent behavior when altering attention scores amongst all layers. Therefrom we derive an explainability method that reveals insights into the black-box correlation of input to output tokens. It also gives some insights into the processing behavior of input tokens throughout the layers of a transformer network. Recent work such as [0], highlight the relevance and novelty of investigating influence functions for transformer interpretability.
>
> [0] Grosse et al. 2023, ‘Studying Large Language Model Generalization with Influence Functions’
>
> Q1
> We again would kindly disagree that AtMan relies on “simple” interventions as we argue above.
>
> Q2
> That is indeed an interesting question that unfortunately is out of scope for this work. In the visual domain, it is natural to ‘look at’ the cosine similarity directly after the embedding layer, in particular as we use CLIP components (c.f. line 199) that are trained with a contrastive loss. We empirically show how it can be utilized to ‘sufficiently suppress’ the input entropy of some concepts, and that measuring those remains stable throughout a large benchmark set.
> In particular, in decoder models, the embeddings of the last layer are not as simple to compare. We assume, as your next question suggests, that you refer to running an embedding model (like BERT) and use those embeds to compute the cosine similarity on the inputs. This indeed would work, and tailor the application to a specific focus (the one BERT is trained with). However one would require an additional inference on a different model to produce those embeddings on the input chunks. In the scope of our work we introduce a XAI method for autoregressive models such as GPT which don’t provide a final output embedding (cf Q3).
>
> Q3
> BERT is an encoder by design, without the causal mask (cf line 165), thus tokens can attend to all other tokens in the attention mechanics. It is usually not applied in practice as a text-generator, but trained with a contrastive loss to generate comparable embeddings. GPT, in contrast, is generative, applying the causal mask, thus tokens can only attend to preceding tokens. This architecture has shown outstanding text-generation capabilities, but due to the masking, the altered meanings of the inner workings remain widely unclear. The resulting embeddings are not as easy to digest.
>
> Q4
> We addressed this quantitatively with attached Figure 17 (‘mask’ row). In the main paper we gave qualitative and quantitative evidence with Fig 3a, Fig 10 and 11. Precision and Recall both severely degenerate.
>
> Q5
> Our intervention receives cosine similarity measures from the first (embedding) layer, and rigorously applies manipulations throughout the network. It is not possible to apply cosine similarity on intermediate layer embeddings out of the box, most likely due to the causal mask. However, we used the initial similarity values and applied the manipulations only in one layer at a time. In the attached Figure 18 we observed an interesting variance, but overall degenerative performance.
>
> We hope we clarified our contribution and resolved your concerns. If you have further questions, please let us know.
>
>
> The authors.

---

> > ### Comment · Reviewer_HHf8 · 2023-08-16
> >
> > Thank you for your detailed response to my questions and concerns, I appreciate the effort. I will be keeping them in mind as we discuss the paper.

---

### Author Rebuttal · Authors · 2023-08-09

Dear All,

thank you for taking the time to review our paper and the valuable feedback. We appreciate the insights and suggestions, which have inspired us to refine and improve our work. In response, we herewith aim to clarify some major concerns that arose among several reviewers

First, we'd like to emphasize that our proposed method is the first XAI technique specifically suited and evaluated for large generative models, that is moreover fit for productive deployment. This key message seems to be misunderstood at times. In particular, to deploy AtMan, it does not require additional workers specifically waiting for interpretability requests, as it would be the case for other methods. This is due to the fact that AtMan only relies on plain forward passes, while gradient-based methods would require a different compiled execution graph, and a different layout among GPUs to ensure enough free memory for the backward pass overhead, optimizer state, and such. Yes, we achieve this by trading memory consumption for run time. However, previous gradient-based methods such as Chefer et al. would not have been deployable at the same hardware cost. Furthermore they require additional workers specifically for XAI requests. To prevent further misunderstandings we updated Figure 5 (see attached Fig. 5).
Efficient deployable XAI methods for generative transformers are especially relevant considering recent stipulations of the EU AI Act, i.e. Article 13, stating that ‘users should be able to “interpret” the system’ and that it ‘requires human oversight measures that can facilitate the “interpretation” of its outputs’.


However, we agree that demonstrating a realistic scenario with the computational cost of AtMan would further clarify the almost negligible additional run time costs.  The attached Fig. 16 compares a practical query (average 165 tokens) explaining an image (144) tokens. Runtimes are measured with parallel execution (pipe-parallel 4, batch size 16). This demonstrates that AtMan is absolutely practical and usable. Specifically, AtMan requires between 1 and 3 total compute seconds, around 1 magnitude longer compared to Chefer. Note that this can still further be divided by the number of available idling workers to further reduce absolute computation time. Each batch of AtMan can be processed entirely parallel.
Further, note that existing perturbation methods are simply not computable at all on large transformer networks due to their large number of trials. Moreover Chefer, even with memory optimizations, fails to scale to 34b with the given memory limit (80GB).


Please further note:

1) We have anonymized the code related to our paper and submitted it to the AC, who will handle further processing and forward it to you.

2) We have incorporated your feedback into the paper to the best of our understanding, enhancing the clarity and improving the overall quality of the submission.

3) We attached Figure 17 demonstrating an exhaustive search on $f$ and $\kappa$ as described in the paper. It can indeed be observed that our proposed subtle modifications generate better explanations than e.g. entirely masking out tokens.


We believe this submission contains enough novel material to better understand the inner workings of transformers, as well as encourages further research, as the reviewers also rightly pointed out plenty-wise, in this understaffed segment of XAI methods on transformers.


Again thank you so much for this fruitful discussion, and kindly find the Figures attached to this post,

the authors of AtMan.

---

### Decision · Program_Chairs · 2023-09-21

**Decision:**

Accept (poster)

**Comment:**

This paper proposes a perturbation method for explaining the prediction of generative Transformer model. The proposed scheme is to manipulate the attention mechanism of the Transformer, so that relevance map of the input can be obtained. The proposed method is memory friendly. Experiments on text and image-text benchmarks demonstrate that the proposed method outperforms gradient-based methods.

The paper has been reviewed by five reviewers. Overall the reviews are positive, except one reviewer HHf8, who gave a rating 3. This reviewer is mainly concerned with novelty. The authors provided a detailed reply on this issue. Reviewer wnDz, who gave a rating 5, is concerned with the inference speed and the choice of tuning parameters. The authors addressed these concerns reasonably. There are other concerns that were not very serious, and were addressed by the rebuttal and discussion.

Overall the proposed method is a useful contribution to explainable AI.